EMBO
*reports*

# Mutual regulation of spermatogenesis-specific Argonaute proteins and Insulin/IGF-1 signaling in aging control

Thomas Liontis [iD] [1,2], Valentina T Pannarale [iD] [1], Andrés R Mansisidor [iD] [3], Sasiru K Pathiranage [iD] [1], Jeeya Y Patel [iD] [1,4] & Alla Grishok [iD] [1,3,5] [✉]

## Abstract

The potential role of small interfering RNAs (siRNAs) produced from double-stranded RNA in aging has not been fully addressed. The networks of genes regulated by siRNAs and their partner Argonaute proteins are best understood in *C. elegans*, a pioneering model of aging and small RNA studies. Here, we describe synergistic lifespan extension of insulin/IGF-1 signaling (IIS) mutant *age-1(hx546)* by *rde-4* or *alg-3; alg-4* deficiencies. By analyzing gene expression and siRNA populations in these IIS and RNAi mutants, we show here that redundant spermatogenesis-specific Argonautes ALG-3 and ALG-4 are capable of regulating IIS, potentially through direct control of the Major Sperm Protein (MSP) genes in the germline. MSPs and MSP domains of some mammalian proteins are secreted and directly inhibit the Eph receptor (EphR). In turn, EphR interacts with and destabilizes PTEN, a major negative regulator of IIS. We show that enhanced MSP expression correlates with EphR mislocalization and elevated PTEN levels in oocytes of *alg-3/4(-)* worms. At the same time, ALG-3/4 expression is regulated by IIS. Thus, we propose mutual regulation of IIS and ALG-3/4 through secreted ligands.

**Keywords** Insulin/IGF-1 Signaling; Argonaute; siRNA; MSP; EphR
**Subject Categories** Metabolism; RNA Biology

## Introduction

The nematode *C. elegans* is an established model organism for aging research (Lapierre and Hansen, 2012). This model also facilitated the pioneering discoveries of microRNA (miRNA) (Lee et al, 1993; Wightman et al, 1993; Pasquinelli et al, 2000; Reinhart et al, 2000) and RNA interference (RNAi) (Fire et al, 1998). Reduction-of-function mutants in the conserved Insulin/Insulin

growth factor-1 Signaling (IIS) pathway components are long-lived due to the activation of transcription factors DAF-16 (FOXO), HSF-1, SKN-1 (NRF), and activation or repression of their targets (Lapierre and Hansen, 2012). This relationship holds in organisms ranging from *C. elegans* to humans (Junnila et al, 2013; Berryman et al, 2008; Bartke and Darcy, 2017; Martins et al, 2016). Importantly, the *FOXO3A* gene is one of only two or three genes that have consistently shown genome-wide association for human longevity (Broer et al, 2015; Deelen et al, 2019). Although the IIS pathway is intensely studied, a complete understanding of how it regulates longevity is lacking due to its impact on hundreds of genes.

The roles of microRNAs in reducing and promoting lifespan have been documented in *C. elegans* (Aalto et al, 2018; Elder and Pasquinelli, 2022) and other organisms (Boehm and Slack, 2006). Moreover, miRNA-binding *C. elegans* Argonaute proteins ALG-1 and ALG-2 (Grishok, 2013) were shown to regulate lifespan oppositely: ALG-1 promotes lifespan, and ALG-2 inhibits it (Aalto et al, 2018). Interestingly, genes regulated by ALG-1 and ALG-2 are enriched in the targets of transcription factor DAF-16/FOXO (Aalto et al, 2018). In contrast to miRNAs, the potential role of endogenous small interfering RNAs (endo-siRNAs) and Argonaute proteins bound to them in aging remains largely unexplored. Currently, there is published evidence supporting the existence of both lifespan-promoting (Cohen-Berkman et al, 2020; Mao et al, 2020; Mansisidor et al, 2011) and lifespan-reducing siRNAs in *C. elegans* (Sebastiani et al, 2009). This study focuses on the role of siRNAs produced by Dicer cleavage of endogenous dsRNA and dependent on the function of the dsRNA-binding protein RDE-4 (Tabara et al, 2002; Parker et al, 2006; Parrish and Fire, 2001). RDE-4 is required for the generation of endogenous small RNAs acting in two distinct pathways, distinguished by the specific Argonaute proteins bound to them: ERGO-1 or ALG-3/4 (Grishok, 2013). RDE-4, along with Argonaute RDE-1, is also a major player in the response of *C. elegans* to exogenous dsRNA (Tabara et al, 2002). ERGO-1 regulates genes in oocytes and embryos, whereas ALG-3 and ALG-4, which are redundant (hereinafter referred to as ALG-3/4), are active in the spermatogenic germline (Han et al,

[1]Department of Biochemistry & Cell Biology, Chobanian & Avedisian School of Medicine, Boston University, 72 East Concord Street, Boston, MA 02118, USA. [2]Graduate Program in Genetics and Genomics, Chobanian & Avedisian School of Medicine, Boston University, Boston, MA 02118, USA. [3]Department of Biochemistry and Molecular Biophysics, Columbia University, New York, NY 10032, USA. [4]Research Science Institute, Massachusetts Institute of Technology, 77 Massachusetts Ave, Cambridge, MA 02139, USA. [5]Genome Science Institute, Boston University, Boston, MA 02118, USA. [✉]E-mail: agrishok@bu.edu

2009; Conine et al, 2010; Corrêa et al, 2010; Seroussi et al, 2023; Vasale et al, 2010).

Little is known about the contribution of proteins expressed in the spermatogenic germline to the regulation of lifespan. Nematode ameba-like crawling sperm differs from that of most animals and uses Major Sperm Proteins (MSPs) in place of the actin cytoskeleton (Bottino et al, 2002). Numerous genes coding for MSPs are expressed in the spermatogenic germline of nematodes. Although most MSPs have a structural role (Italiano et al, 1996), some are secreted and inhibit Eph receptor (EphR) signaling in oocytes, in a mechanism that promotes oocyte maturation (Miller et al, 2001, 2003; Cheng et al, 2008). Proteins containing MSP-like domains are present in many organisms, including humans. The most notable is the mammalian VAP (VAMP)-associated protein, VAPB, which is a conserved ubiquitously expressed ER protein (Nishimura et al, 1999), mutated in Amyotrophic Lateral Sclerosis (ALS) patients (Nishimura et al, 2004). Intriguingly, expression of several germline MSPs has been positively correlated with extended lifespan through elevated expression in long-lived Insulin/IGF-1 receptor mutant *daf-2* (Gao et al, 2018) and dietary restriction *eat-2* mutant (Heestand et al, 2013; Gao et al, 2018) models and reduced expression in short-lived miRNA mutant *mir-71* (Inukai et al, 2018). Moreover, downregulation of the MSP gene *dct-9* suppressed the extended lifespan of the *daf-2* mutant (Pinkston-Gosse and Kenyon, 2007).

Here, we find synergistic extensions of lifespan and healthspan from disruption of the RDE-4/ALG-3/4 and IIS pathways. Since MSP genes correlated with extended lifespan are repressed by the RDE-4/ALG-3/4 pathway and show elevated expression in the *rde-4* and IIS pathway mutants, we tested the possibility that molecular and cellular changes consistent with enhanced MSP secretion occur in *alg-3; alg-4* mutant worms. Indeed, we found evidence that ALG-3/4 deficiency non-cell-autonomously leads to VAB-1 inhibition and elevated DAF-18 levels, which should cause DAF-16 activation. Unexpectedly, we also found that IIS promotes *alg-3/4* expression in the spermatogenic germline. Overall, our results highlight the potential of endogenous RNAi components in modulating signaling pathways in a non-autonomous manner, and, likewise, suggest that IIS affects the expression of key proteins, which execute siRNA-based gene regulation.

## Results

### Disruption of the RDE-4/ALG-3/4 pathway delays aging synergistically with mildly reduced IIS

The RDE-4 dsRNA-binding protein cooperates with ribonuclease Dicer in cleaving dsRNA molecules and producing siRNAs (Tabara et al, 2002; Parrish and Fire, 2001; Parker et al, 2006). In the exogenous RNAi pathway, RDE-4 is essential and generates siRNAs bound to Argonaute RDE-1 (Yigit et al, 2006). In endogenous RNAi pathways, RDE-4 works in two Argonaute branches: ERGO-1 and ALG-3/4 (Grishok, 2013). Importantly, whereas Dicer cleaves both long dsRNA and short miRNA precursors, RDE-4 does not participate in miRNA maturation (Grishok, 2013). We aimed to determine whether the specific loss of dsRNA-derived endo-siRNAs affects *C. elegans* lifespan by taking advantage of the null *rde-4* mutant *rde-4(ne301)* (Tabara

et al, 2002). We used strains outcrossed six times to the N2 wild type (WT) strain to mitigate effects from background mutations. We observed a mild and variable extension of lifespan in *rde-4* mutants (+ 8% mean lifespan (MLS)), where pooling of several biological replicates was required to detect the significant increase in lifespan (Fig. 1A, all lifespan replicate data are compiled in Dataset EV1). Since genetic interactions between the RNAi and IIS pathways have been observed in several studies (Mansisidor et al, 2011; Wang and Ruvkun, 2004; Simon et al, 2014), we chose to assess the interaction between endogenous RNAi and IIS in lifespan regulation. We used the *age-1(hx546)* mutant background, which results in longevity due to a moderate decrease in IIS and activation of DAF-16/FOXO (Friedman and Johnson, 1988; Klass, 1983; Ayyadevara et al, 2008). Interestingly, the lifespan extension by the *rde-4* mutation was strongly enhanced and consistent in long-lived *age-1* mutants (+ 25% MLS in *age-1* mutants and an additional +16% MLS in *age-1; rde-4* mutants, compared to WT) (Fig. 1B, Dataset EV1).

To further understand the nature of this synergistic interaction between *rde-4* and *age-1*, we performed mRNA- (Dataset EV2) and small RNA- (Dataset EV3) sequencing using WT and *rde-4*, *age-1*, and *age-1; rde-4* mutant strains. Our small RNA-sequencing protocol enriches for Dicer products with a 5' monophosphate, notably primary endo-siRNAs. As expected, we found a correlation between siRNA depletion and its target mRNA elevation in *rde-4* mutants consistent with negative gene regulation by siRNAs (Fig. 1C, Dataset EV4). Surprisingly, this effect was even more pronounced in the *age-1(hx546)* mutant background (Fig. 1D). In *age-1(hx546)* mutants, we also detected a positive correlation between siRNA and mRNA levels (Fig. 1E, Datasets EV4–EV6). Importantly, we observed elevated expression of genes repressed by RDE-4 in *age-1(hx546)* (Figs. 1F and EV1A), which suggests defects of RNAi-based gene silencing in this IIS mutant. Specifically, genes repressed by ALG-3/4-bound siRNAs (Conine et al, 2010) were elevated in both *rde-4* and *age-1* mutants (Fig. 1G, Datasets EV5, EV6). Importantly, the majority of genes targeted by RDE-4-dependent siRNAs overlap with ALG-3/4 targets (Seroussi et al, 2023) (Dataset EV3). Also, genes corresponding to RDE-1-dependent small RNAs ("RDE-1 targets") (Seroussi et al, 2023) were upregulated in *age-1(hx546)* (Fig. EV1B). In addition to gene silencing, ALG-3/4-associated endo-siRNAs can promote the expression of their gene targets ("positive targets") notably at 25 °C (Conine et al, 2013). This effect appears to be enhanced by *age-1(hx546)* (Figs. 1G and EV1C). Thus, the *age-1(hx546)* background modulates the gene expression of ALG-3/4 target genes.

To determine whether impairment in the ALG-3/4 or RDE-1 pathways contributes to the synergistic longevity of *age-1; rde-4* double mutants, we measured the lifespan of the corresponding mutants in the WT and *age-1(hx546)* backgrounds. Excitingly, the *alg-3; alg-4* mutations strongly and consistently extended lifespan in the *age-1(hx546)* background (+33% MLS in *age-1* mutants and an additional +20% MLS in *age-1; alg-3; alg-4* mutants, compared to WT) (Fig. 1H, Dataset EV1), phenocopying the nonlinear, synergistic effect of *rde-4(-)* on *age-1(hx546)* longevity (Fig. 1A,B). Indeed, the lifespan extension by *alg-3(-); alg-4(-)* in a WT background was mild (+ 5% MLS) and variable (Fig. 1H, Dataset EV1). Meanwhile, the *rde-1* mutation resulted in a mild and variable extension of lifespan in both WT (+ 5% MLS) and

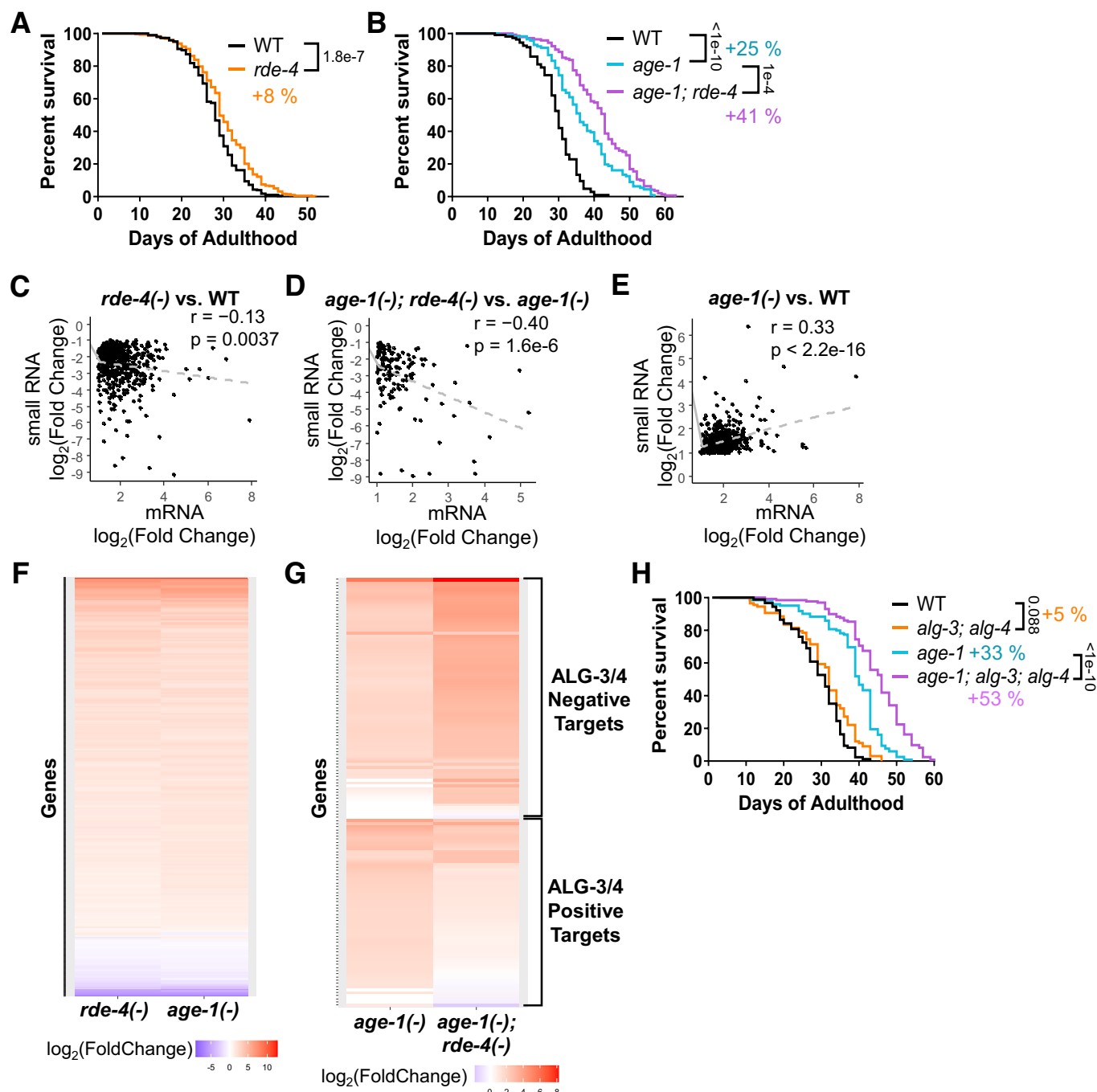

*age-1(hx546)* backgrounds (+ 25% MLS in *age-1* mutants and an additional +7.5% MLS in *age-1; rde-1* mutants, compared to WT), an effect that appears to be linear (Fig. EV1D, Dataset EV1). Moreover, we measured the lifespan of male worms. Although *age-1(hx546)* lived longer compared to WT, the addition of *alg-3(-); alg-4(-)* did not cause lifespan extension in males (Fig. EV1E). ALG-3/4 function may be more critical for male physiology compared to hermaphrodites or pro-longevity signaling in *age-1; alg-3; alg-4* mutants may require components present in hermaphrodites but not males.

To further connect the *age-1(hx546); rde-4(-)* life extension to ALG-3/4 pathway deficiency, we conducted lifespan experiments using *nrde-3* mutants deficient in the downstream steps of the ERGO-1 pathway (Guang et al, 2008), since ERGO-1 also binds RDE-4-dependent siRNAs (Grishok, 2013). The *age-1; nrde-3* double mutant worms lived shorter compared to *age-1* alone (Fig. EV1F), consistent with the reported pro-longevity function of the ERGO-1 pathway (Cohen-Berkman et al, 2020).

We conclude that decreased activity of the ALG-3/4 pathway in *rde-4* and *alg-3/4* mutant hermaphrodites results in lifespan

Figure 1. Disruption of RNAi components RDE-4 and ALG-3/4 extends lifespan synergistically with moderately reduced IIS.

(A, B) Survival curves showing that the *rde-4(ne301)* mutation extends lifespan mildly in the wild-type (WT) background (+ 8% mean lifespan (MLS), *n* = 473 and 564 animals) (A) and strongly in the long-lived *age-1(hx546)* (+25% MLS, *n* = 152 animals) background (+ 16% additional MLS in *age-1; rde-4* mutants, *n* = 152 animals) (B). (C–E) Scatter plots of differentially expressed mRNAs and antisense small RNAs with |log$_2$(fold change)| > 1. Anticorrelations between downregulated antisense small RNAs and upregulated mRNAs in *rde-4(ne301)* compared to WT (*n* = 482 genes) (C) and in *age-1(hx546); rde-4(ne301)* compared to *age-1(hx546)* (*n* = 138 genes) (D) suggest RDE-4-dependent siRNAs negatively regulate genes. (E) The correlation for upregulated antisense small RNAs and upregulated mRNAs in *age-1(hx546)* compared to WT (*n* = 687 genes) suggests RDE-4-dependent siRNAs can also positively regulate genes, specifically in the *age-1(hx546)* background. (F, G) Heatmaps showing the log$_2$(fold change) of genes relative to WT. The numerous commonly differentially expressed genes in *rde-4(ne301)* and *age-1(hx546)* tend to vary in the same direction with a similar magnitude of change (F). ALG-3/4 targets (Conine et al, 2013) were largely upregulated in *age-1(hx546)* mutants, with negatively-regulated targets strongly derepressed and positively-regulated targets partially suppressed in *age-1; rde-4* mutants (G). Only genes differentially expressed in *age-1; rde-4* mutants compared to *age-1(hx546)* that showed a change in ALG-3/4 target expression are shown. Genes with nonsignificant changes are simplified to log$_2$(fold change) = 0. (H) Survival curves showing that the *alg-3(tm1155); alg-4(ok1041)* mutations mildly extend lifespan in WT background (+ 5% MLS, *n* = 137 and 150 animals) and strongly extend it in *age-1(hx546)* background (+ 20% additional MLS in *age-1; alg-3; alg-4* mutants, *n* = 139 and 144 animals). All percentage MLS changes are relative to WT. Lifespan experiments (A, B, H) were independently replicated 8, 3, and 2 times, respectively. Survival curves include pooled data from all experimental replicates and are compared by the log-rank test. The Pearson correlation test was used for (C–E). Source data are available online for this figure.

extension that is strongly enhanced in the moderate IIS mutant *age-1(hx546)*.

## Correlation between the enhanced lifespan and healthspan in *age-1; alg-3; alg-4* mutants

As worms age, they become increasingly frail and eventually cease to move spontaneously, despite being alive and moving once prodded. While measuring the lifespan of *age-1; alg-3; alg-4* triple mutants, we noticed, despite being blinded to the genotype, that the worms appeared healthier at time points before their longevity became obvious (Fig. EV1G). To quantify this healthspan effect, we repeated the lifespan assay while counting the number of worms that were both alive and spontaneously moving. Consistent with previous research (Duhon and Johnson, 1995), we found that *age-1* mutants exhibited a mildly prolonged healthspan, which was statistically significant at only one of the tested time points, day 34 of adulthood (Day 34) (Fig. 2A). Strikingly, *age-1; alg-3; alg-4* triple mutants had an even further extended healthspan than *age-1(hx546)*, at several timepoints (Days 32, 39, and 46), which means these worms not only live longer but also have more preserved movement throughout their life (Fig. 2B).

To assess the reproductive health of our strains, we measured their brood size. Consistent with previous research (Seroussi et al, 2023; Blanchard et al, 2011), *rde-1* and *rde-4* mutants had fewer progeny that developed normally, and this was not exacerbated in *age-1* mutants (Fig. 2C). The *alg-3; alg-4* mutants have defective sperm leading to a strongly decreased brood size at 20 °C and sterility at 25 °C, which are essentially fully rescued by WT sperm (Conine et al, 2010). Interestingly, the small brood size of *alg-3; alg-4* mutants at 20 °C was significantly worsened in the *age-1(hx546)* background (Fig. 2D). To assess whether this combinatorial decrease in brood size was also caused by a sperm defect, we tested whether the near-sterility of *age-1(hx546); rde-4(-)* and full sterility of *age-1(hx546); alg-3(-); alg-4(-)* hermaphrodites at 25 °C could be rescued by *age-1(hx546)* male sperm (Fig. 2E). We found that in both cases, fertility could indeed be restored with *age-1(hx546)* sperm. However, the extent of the brood size rescue by *age-1(hx546)* sperm was largely incomplete in *age-1(hx546); alg-3(-); alg-4(-)* hermaphrodites, suggesting they may also have an oocyte defect (Fig. 2E).

Finally, we measured the incidence of deaths considered "unnatural" or "accidental" in the tested strains over their lifespan.

While "natural" deaths of aging occur slowly and are often accompanied by a gradual decrease in spontaneous movement, "accidental" deaths are defined by sudden deaths following an incident, such as internal hatching of progeny or leaking of the intestine, preceded by vulval bursting. These "accidental" deaths are typically right-censored in lifespan analyses (Cornwell and Samuelson, 2020; Park et al, 2017) but may provide additional insights into the health and physiological state of different worm strains. The *age-1; rde-4* and *alg-3; alg-4* mutants displayed a decrease in deaths caused by internal hatching of their progeny (Fig. EV2A,B), which may be explained by their very low brood size (Fig. 2C,D). Interestingly, the *rde-1* and *rde-4* mutations decreased the incidence of intestinal leakage-related deaths, specifically in the *age-1(hx546)* background (Fig. EV2C). Intriguingly, *alg-3; alg-4* mutants showed a strong increase in deaths by leaking of the intestine (25% of worms compared to 10% in WT), which was fully suppressed by *age-1(hx546)* (Fig. EV2D). These results suggest additional genetic interactions between endogenous RNAi pathways and IIS, where the *age-1(hx546)* background suppresses "unnatural" deaths caused by leaking of the gut in *alg-3; alg-4* mutants.

Overall, we found that the combined impairments of *alg-3*, *alg-4*, and *age-1* resulted in a prolonged healthspan and lifespan.

## ALG-3/4 inhibition correlates with lifespan extension in *daf-2(e1370)*

Next, we tested whether the lifespan extension of *age-1(hx546); rde-4(-)* was dependent on DAF-16 and found that it was fully suppressed by the *daf-16(mgDf50)* mutation (Fig. 3A). Interestingly, the highly extended lifespan of *daf-2(e1370)* mutants, which have a stronger decrease in IIS and activation of DAF-16 compared to *age-1(hx546)* (Dorman et al, 1995; Henderson and Johnson, 2001; Gottlieb and Ruvkun, 1994), was not further extended by the loss of RDE-4 (Fig. 3B,C). Since the lifespan of *daf-2(e1370)* worms is strongly extended by germline ablation (Hsin and Kenyon, 1999), our results suggest that the effect of *rde-4(-)* on longevity is distinct from germline loss. Moreover, these results indicate that disruption of RDE-4-dependent endogenous RNAi pathways can extend the lifespan of *age-1(hx546)* by further activating DAF-16 (Fig. 3C, right). In longer-lived *daf-2* mutants, however, this enhancement of DAF-16 activity may already be occurring such that the *rde-4* mutation does not affect lifespan (Fig. 3C, middle).

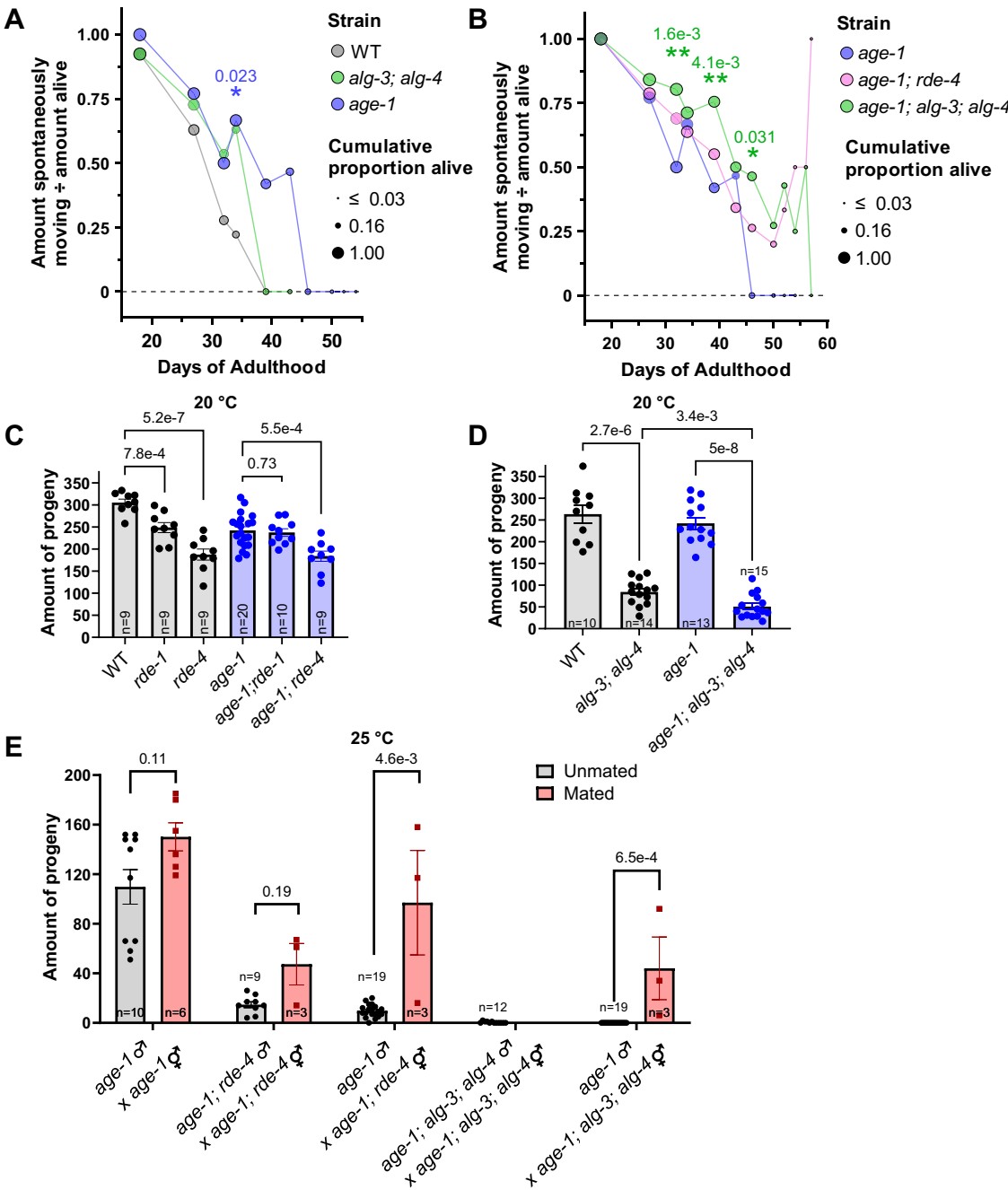

**Figure 2. Disruption of RNAi in *age-1* mutants extends healthspan at the expense of fertility.**

(A, B) The number of worms exhibiting spontaneous movement ("amount spontaneously moving") among those alive decreases with age. At Day 34 of adulthood, a larger proportion of live worms exhibit spontaneous movement in *age-1(hx546)* than WT ($n = 45$ and 9 animals) (A). A larger proportion of live worms exhibit spontaneous movement in *age-1(hx546); alg-3(tm1155); alg-4(ok1041)* than *age-1(hx546)* at Day 32 ($n = 61$ and 46 animals), Day 39 ($n = 49$ and 31 animals), and Day 46 ($n = 28$ and 7 animals) (B). *P* values of these comparisons are indicated above asterisks. (C, D) Brood size in the WT or *age-1(hx546)* background of RNAi mutants *rde-1(ne219)*, *rde-4(ne301)* (C), and *alg-3(tm1155); alg-4(ok1041)* (D). (E) Brood size at 25 °C in the *age-1(hx546)* background in progeny resulting from crossing *age-1(hx546)* control males with hermaphrodite RNAi mutants or RNAi mutant males with isogenic hermaphrodites. Crosses with successful mating indicated in red bars, unsuccessful crosses (self-fertilization) in gray bars. Error bars are s.e.m. Fisher's exact test was used for (A) and (B). Bars represent means and are compared with unpaired two-tailed tests: t-tests in (C–E) and Mann–Whitney tests in (D, E) (see Methods for decisions between t and Mann–Whitney tests, as well as corrections). Sample sizes for (C–E) are indicated at the bottom of graphs and always represent the number of biological replicates (animals) in all figures. ns: not significant for all figures. Source data are available online for this figure.

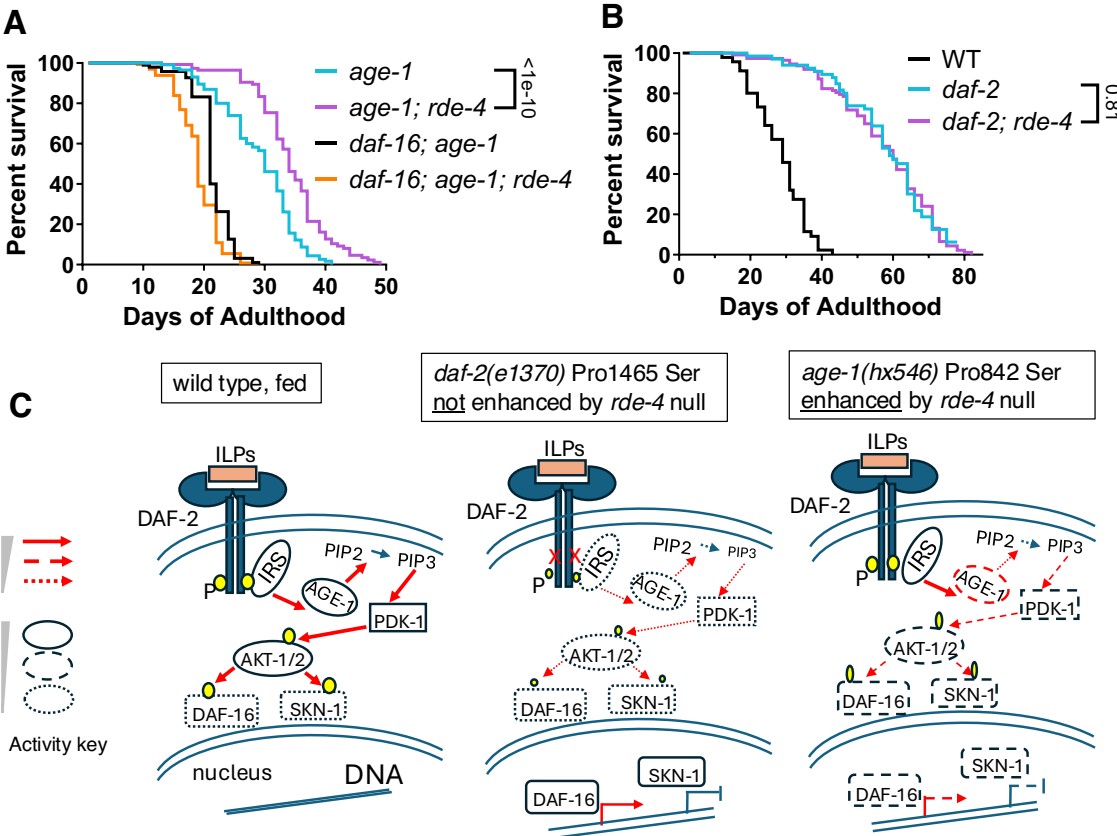

**Figure 3. Longevity by loss of *rde-4* in *age-1(hx546)* depends on DAF-16 and does not occur in *daf-2(e1370)*, where IIS is strongly reduced.**

(A) Survival curves showing that lifespan extension from the *age-1(hx546); rde-4(ne301)* mutations (*n* = 143 and 138 animals) is suppressed in a *daf-16(mgDf50)* background (*n* = 132 and 145 animals). Pooled data from 3 independent experiments. (B) Survival curve showing that *rde-4(ne301)* does not extend lifespan in a *daf-2(e1370)* background (*p* = 0.81, *n* = 140 and 167 animals). Pooled data from 2 independent experiments. Comparisons in (A, B) were done with the log-rank test. (C) Schematic illustrating genetic interactions between the partial loss-of-function *daf-2(e1370)* and *age-1(hx546)* mutants and *rde-4* null mutant. *daf-2(e1370)* is stronger than *age-1(hx546)*. Source data are available online for this figure.

Consistent with the possibility of impaired endogenous RNAi in *daf-2(e1370)*, we observed that genes upregulated in *daf-2(e1370)* compared to WT (Zullo et al, 2019) and in *age-1; rde-4* compared to *age-1* mutants significantly overlapped (Fig. EV3A), as is the case for downregulated genes in both comparisons (Fig. EV3B). Excitingly, *alg-3* and *alg-4* were found to be strongly downregulated in *daf-2(e1370)* (Zullo et al, 2019) (Fig. EV3C). For technical reasons, we could not observe possible increased nuclear localization of endogenously tagged DAF-16::GFP in *age-1(hx546); rde-4(-)* (Fig. EV3D,E), as it is very sensitive to external conditions and already resides in the nucleus of animals taken for imaging.

Next, we evaluated *alg-3* and *alg-4* mRNA expression in WT, *daf-2(e1370)*, *daf-16(mgDf50)*, and double *daf-2; daf-16* mutant worms. We performed a time-series quantitative PCR (qPCR) experiment including young L4, late L4, young adult, and D2 adult stages (Fig. EV4A,B). We found that *alg-3* mRNA was significantly decreased in *daf-2(e1370)* mutants specifically at the late L4 stage, coinciding with the stage when *alg-3* RNA levels peak (Fig. EV4A). Importantly, this regulation of *alg-3* mRNA in *daf-2(e1370)* mutants was dependent on DAF-16 (Fig. EV4A). Changes in *alg-4* followed similar trends, but were not statistically significant

(Fig. EV4B). We also used an endogenously-tagged GFP::ALG-3 strain (Charlesworth et al, 2021) to assess ALG-3 protein levels in these strains over further subdivided developmental stages (Fig. 4A). ALG-3 protein levels were initially lower in both *daf-2* and *daf-2; daf-16* mutant strains during young and moderate L4 stages (Fig. 4A,B), which may have been caused by their slower growth. We observed a DAF-16-dependent decrease in ALG-3 protein levels in *daf-2* mutants beginning at the late L4 stage (Fig. 4A,C), where ALG-3 is expressed in spermatocytes. This DAF-16-dependent decrease in ALG-3 was particularly striking in young adult spermatocytes (Fig. 4A,D) and residual bodies (Fig. EV4C), and persisted in residual bodies of moderate Day 1 adults (Fig. 4A,E). Male *daf-2* mutant young adults also showed a decrease in spermatocyte ALG-3 protein levels (Fig. EV4D). Expression of *alg-3* and *alg-4* in hermaphrodites was subsequently depleted in all strains by day 2 of adulthood (Figs. 4A and EV4A,B). In contrast to *daf-2* mutants, *age-1(hx546)* mutants displayed a very mild decrease in ALG-3 levels (Fig. EV4E).

Overall, we found that ALG-3 is already repressed in a strong reduction of IIS background, *daf-2(e1370)*, and that impairing RDE-4-dependent endogenous RNAi, which is predominantly

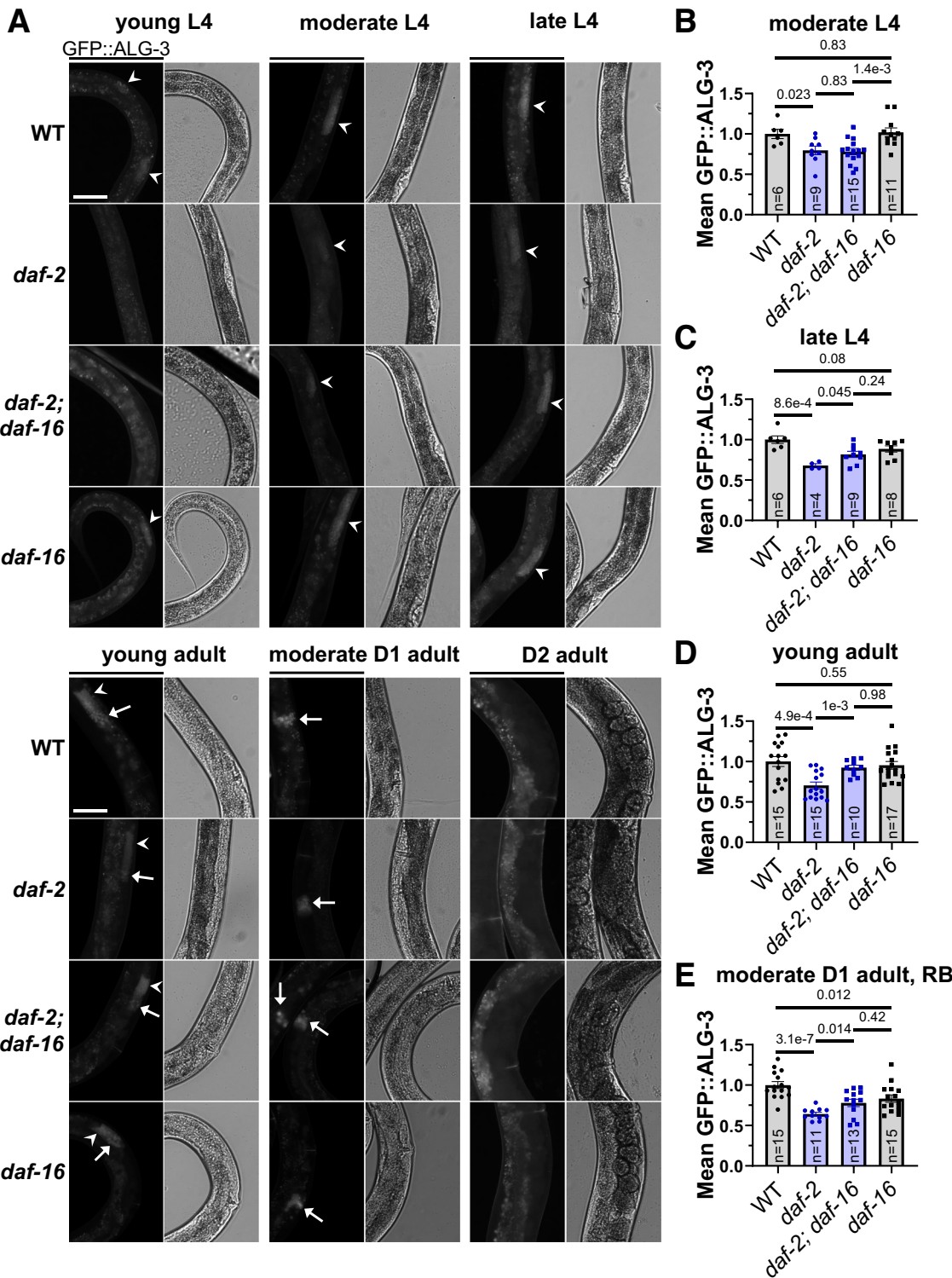

**Figure 4. Strong impairment of IIS results in decreased ALG-3/4 levels dependent on DAF-16.**

(A) Representative images in young L4, moderate L4, late L4, early young adult, moderate Day 1 (D1) adult, and Day 2 (D2) adult stages of endogenously-tagged GFP::3xFLAG::ALG-3 (GFP::ALG-3) expressed in spermatocytes (arrowheads) and/or residual bodies (RB, arrows) in WT, *daf-2(e1370)*, *daf-16(mgDf50)*, and double *daf-2; daf-16* mutant worms. Exposure was kept constant for all strains of a given stage, but not between stages. Scale bars: 50 μm. (B–E) Quantification of GFP::ALG-3 in spermatocytes of moderate L4s (B), late L4s (C), early young adults (D), and RB in moderate D1 adults (E). Error bars are s.e.m. for (B–E). Sample sizes for (B–E) are indicated in the graphs and represent the number of biological replicates (animals). Bars were compared with unpaired two-tailed tests: t-tests in (B–E) and Mann–Whitney test in (D) (see Methods for decisions between t and Mann–Whitney tests, as well as corrections). Source data are available online for this figure.

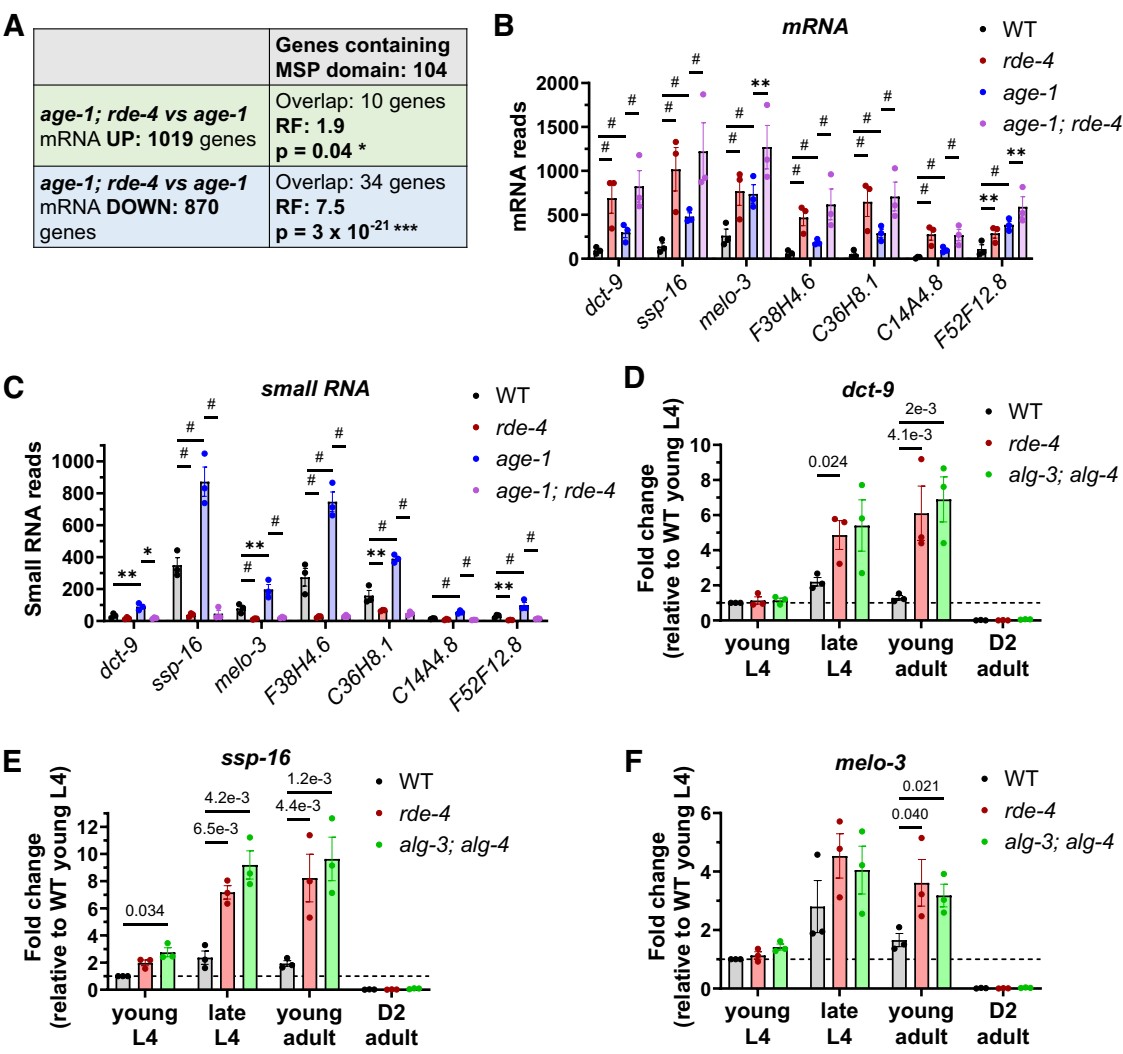

**Figure 5. Specific MSP domain-containing genes are repressed by RDE-4 and ALG-3/4.**

(A) Table displaying the enrichment (Fisher's exact test) for genes containing the major sperm protein (MSP) domain among genes upregulated or downregulated in *age-1(hx546); rde-4(ne301)* relative to *age-1(hx546)*. (B) DESeq2-normalized mRNA reads of genes containing the MSP domain that are significantly upregulated in *rde-4(ne301)*, *age-1(hx546)*, and *age-1(hx546); rde-4(ne301)*. (C) Raw small RNA reads of the same genes. (D–F) Time-series qPCRs during young L4, late L4, early young adult, and D2 adult stages of highly expressed MSP-containing genes *dct-9* (D), *ssp-16* (E), and *melo-3* (F). Statistical analyses in (B, C) are based on full RNA-sequencing Datasets EV2, EV3 (in which the exact p values are listed) that utilize Wald tests accompanied by complex statistical procedures in the DESeq2 algorithm. Sample size is n = 3 independent worm populations and error bars are s.e.m. (B–F). Comparisons in (D–F) used unpaired two-tailed tests: t-tests in (D–F) and Mann-Whitney tests in (D, F). *p < 0.05. **p < 0.01. *** or #p < 0.001. Source data are available online for this figure.

ALG-3/4-driven (Dataset EV3) (Seroussi et al, 2023), does not extend the lifespan of *daf-2(e1370)*. Consistently, ALG-3/4 depletion results in the extended longevity of a relatively weak IIS mutant, *age-1(hx546)*, where ALG-3 is not strongly repressed. Thus, our results define *alg-3* and *alg-4* as anti-longevity genes upregulated by the DAF-2 signaling pathway. Our findings are consistent with the possibility of direct repression of these genes by DAF-16 in the *daf-2(e1370)* background. Indeed, bioinformatic analyses identified a putative DAF-16 binding site in the *alg-3* promoter (Fig. EV5). Moreover, the existence of conserved potential SKN-1 binding sites at the promoters of both *alg-3* and *alg-4* (Fig. EV5) further supports the possibility of direct repression of these genes by IIS-regulated transcription factors.

## Expression of specific MSPs repressed by ALG-3/4 correlates with lifespan extension

Genes containing the Major Sperm Protein (MSP) domain were significantly enriched among the genes misregulated in *age-1(hx546); rde-4(-)* compared to *age-1(hx546)* (Fig. 5A). Given that ALG-3/4 are important for spermatogenesis and spermiogenesis (Conine et al, 2010), and that *age-1(hx546); rde-4(-)* has a reduced brood size rescued by *age-1(hx546)* sperm (Fig. 2E), the apparent defect in sperm in *age-1(hx546); rde-4(-)* could explain the downregulation of many MSP genes. However, the enrichment for the upregulation of some MSP genes was unanticipated (Fig. 5A).

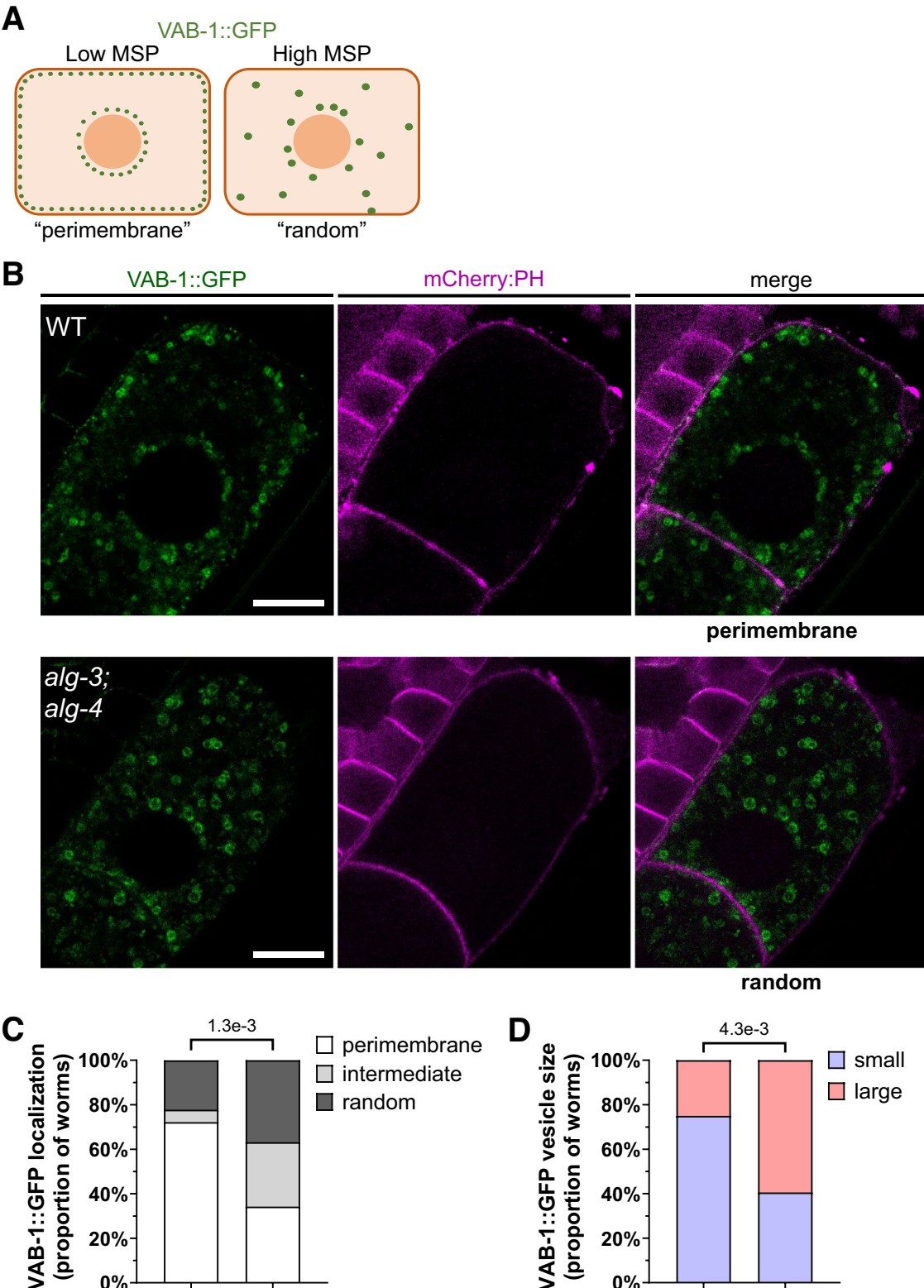

A VAB-1::GFP
Low MSP High MSP

"perimembrane" "random"

B VAB-1::GFP mCherry:PH merge

WT

perimembrane

*alg-3; alg-4*

random

**C**

VAB-1::GFP localization (proportion of worms)

1.3e-3

☐ perimembrane
▨ intermediate
■ random

WT *alg-3; alg-4*

**D**

VAB-1::GFP vesicle size (proportion of worms)

4.3e-3

☐ small
☐ large

WT *alg-3; alg-4*

**Figure 6. ALG-3/4 promotes VAB-1 trafficking.**

(A) Cartoon of expected *pie-1p::VAB-1::GFP* (VAB-1::GFP) interior vesicle (green circle) localization in the −1 oocyte under low or absent MSP, where smaller VAB-1::GFP vesicles predominantly surround (but do not overlap with) the cell membrane ("perimembrane"), in contrast to high MSP levels, where larger VAB-1::GFP vesicles exhibit a more random localization pattern ("random"). The large orange circle in the center is the oocyte nucleus. (B) Representative VAB-1::GFP confocal microscopy images of adult −1 oocytes with "perimembrane" localization in WT and "random" localization in *alg-3(tm1155); alg-4(ok1041)* mutants. Images of all possible localization phenotypes for both strains are shown in Fig. EV6. The *pie-1p::mCherry:PH^{PLC6PH}* transgene (mCherry:PH, pseudocolored magenta) is expressed in the membrane (Kachur et al, 2008). Scale bars: 10 μm. (C) Proportion of worms with perimembrane, intermediate, or random localization of VAB-1::GFP. There is a significant decrease in the proportion of worms with perimembrane VAB-1::GFP vesicle localization compared to pooled random and intermediate localizations ($p = 0.0013$, shown on graph, $n = 36$ and 38 animals) or random localization only ($p = 0.0321$, $n = 27$ and 34 animals) in *alg-3; alg-4* mutants compared to WT. (D) Large VAB-1::GFP vesicles are more often present in *alg-3; alg-4* mutants than in WT ($p = 0.0043$, $n = 36$ and 37 animals). Fisher's exact test was performed in (C, D) from pooled data of over 3 independent experiments. Source data are available online for this figure.

Specifically, seven MSP genes with reasonable expression levels in WT were found to be upregulated in all studied mutant conditions (Fig. 5B). Consistently, the levels of siRNAs antisense to these genes were reduced by *rde-4* loss of function (Fig. 5C). Also, several studies reported the upregulation of these MSPs in endogenous RNAi mutants, such as *rrf-3*, *eri-1*, *dcr-1*, *mut-16*, and *hrde-2* (Asikainen et al, 2007; Manage et al, 2020; Chen and Phillips, 2024; Welker et al, 2007). Through database annotations, we determined that expression of these seven genes was elevated in *daf-2* (Gao et al, 2018) and *eat-2* (Heestand et al, 2013; Gao et al, 2018) long-lived mutants and reduced in the short-lived *mir-71* mutant (Inukai et al, 2018). Moreover, downregulation of at least one MSP from the list, *dct-9*, was implicated in suppression of *daf-2* mutant phenotypes, such as extended lifespan and tumorous germline (Pinkston-Gosse and Kenyon, 2007). Using RT-qPCR, we confirmed the elevated expression of *dct-9*, *ssp-16*, and *melo-3* mRNAs at the late L4 and/or young adult stages in *rde-4* and *alg-3; alg-4* mutants (Fig. 5D–F).

MSPs can be secreted from the spermatogenic germline and disrupt signaling through EphR (Miller et al, 2001, 2003; Cheng et al, 2008). Given the published associations between specific MSPs and lifespan extension (Heestand et al, 2013; Gao et al, 2018), and elevated expression of these same MSPs in *rde-4* and *alg-3/4* mutant conditions that extend the lifespan of a weak IIS mutant, we propose that RDE-4/ALG-3/4 may regulate lifespan by controlling the level of secreted MSPs.

## ALG-3/4 promotes VAB-1/EphR trafficking and inhibits DAF-18/PTEN expression in oocytes

In addition to their structural role, MSP proteins mediate extracellular signaling from sperm to oocyte (Miller et al, 2001, 2003). In particular, MSPs bind to and inhibit the Eph receptor, VAB-1, in oocytes to promote oocyte maturation, while also acting on gonadal sheath cells to promote contraction, thereby promoting ovulation and fertilization (McCarter et al, 1999; Miller et al, 2001, 2003). Sperm MSPs also disrupt the trafficking of VAB-1 in the mature oocyte, where the loss of MSPs causes smaller VAB-1 vesicles that predominantly surround (but do not overlap with) the cell membrane (Cheng et al, 2008) ("perimembrane" in Fig. 6A, left). Meanwhile, the presence of MSPs causes a more random localization pattern enriched for larger interior vesicles ("random") (Cheng et al, 2008; Hang et al, 2008) (Fig. 6A, right). This may be, in part, caused by an overall reorganization of oocyte microtubules in response to MSPs (Harris et al, 2006).

To determine whether the derepressed MSP proteins in *alg-3(-); alg-4(-)* inhibit VAB-1, we assessed the localization of VAB-1 vesicles in mature oocytes using a strain expressing VAB-1::GFP

specifically in the germline (Cheng et al, 2008) (Figs. 6B and EV6). We found that the proportion of *alg-3(-); alg-4(-)* worms showing "perimembrane" VAB-1 expression in the mature oocyte was approximately half of that of WT, while 66% more *alg-3(-); alg-4(-)* worms showed "random" VAB-1 expression compared to WT (Fig. 6C). These results suggest that VAB-1 trafficking is impaired in *alg-3; alg-4* mutants. In addition, the proportion of worms with larger VAB-1 vesicles was significantly increased in *alg-3(-); alg-4(-)* (Fig. 6D).

Importantly, disruption of *vab-1* causes an increase in DAF-18/PTEN protein levels and extends lifespan in worms devoid of self-sperm (Brisbin et al, 2009). Using an endogenously-tagged mNeonGreen::DAF-18 strain (Huang et al, 2021), we found that the *alg-3; alg-4* mutations caused an over 2-fold increase in DAF-18 in the mature oocyte in both the WT and *age-1(hx546)* backgrounds (Fig. 7A,B). The increase in DAF-18 protein levels did not occur in response to the defective paternal sperm of *alg-3(-); alg-4(-)* mutants, as the offspring of *alg-3(-); alg-4(-)* males crossed to WT hermaphrodites did not show increased DAF-18 levels (Fig. 7C). Our observed effects on VAB-1 trafficking and DAF-18 levels (Figs. 6, 7) are consistent with the possibility of increased MSP secretion and activity in *alg-3(-); alg-4(-)* (Cheng et al, 2008; Brisbin et al, 2009; Miller et al, 2003; Hang et al, 2008). Importantly, the upregulation of DAF-18 was also reported in *daf-2(e1370)* (Liu et al, 2014), which may be in part due to the downregulation of *alg-3* and *alg-4* (Figs. 4, 8, EV3C and EV4A,B). DAF-18 over-expression extends lifespan (Brisbin et al, 2009; Solari et al, 2005), and DAF-18 rescue in single tissues in *daf-2; daf-18* mutants was shown to extend lifespan regardless of the tissue, although germline-specific over-expression was not tested (Masse et al, 2005). Nevertheless, the upregulation of DAF-18 by *alg-3(-); alg-4(-)* (Fig. 7A,B) may be sufficient for longevity, with a synergistic effect in the *age-1(hx546)* background, due to both DAF-18 and AGE-1 regulating PI phosphorylation (Fig. EV7) (Solari et al, 2005; Ogg and Ruvkun, 1998).

We propose that the elevation of DAF-18 levels observed in *alg-3(-); alg-4(-)* oocytes might also happen in somatic tissues critical for lifespan extension by IIS mutants (Fig. 8). This would further limit IIS activity in the weak *age-1(hx546)* mutant, extending its lifespan and revealing the contribution of the ALG-3/4 RNAi pathway to longevity restriction.

## Discussion

Discoveries of the systemic nature of RNAi in *C. elegans* (Fire et al, 1998), multigenerational inheritance of small RNAs in worms and flies (Ishizu et al, 2012; Frolows and Ashe, 2021), and secretion/uptake of small RNAs via extracellular vesicles in mammals (Das

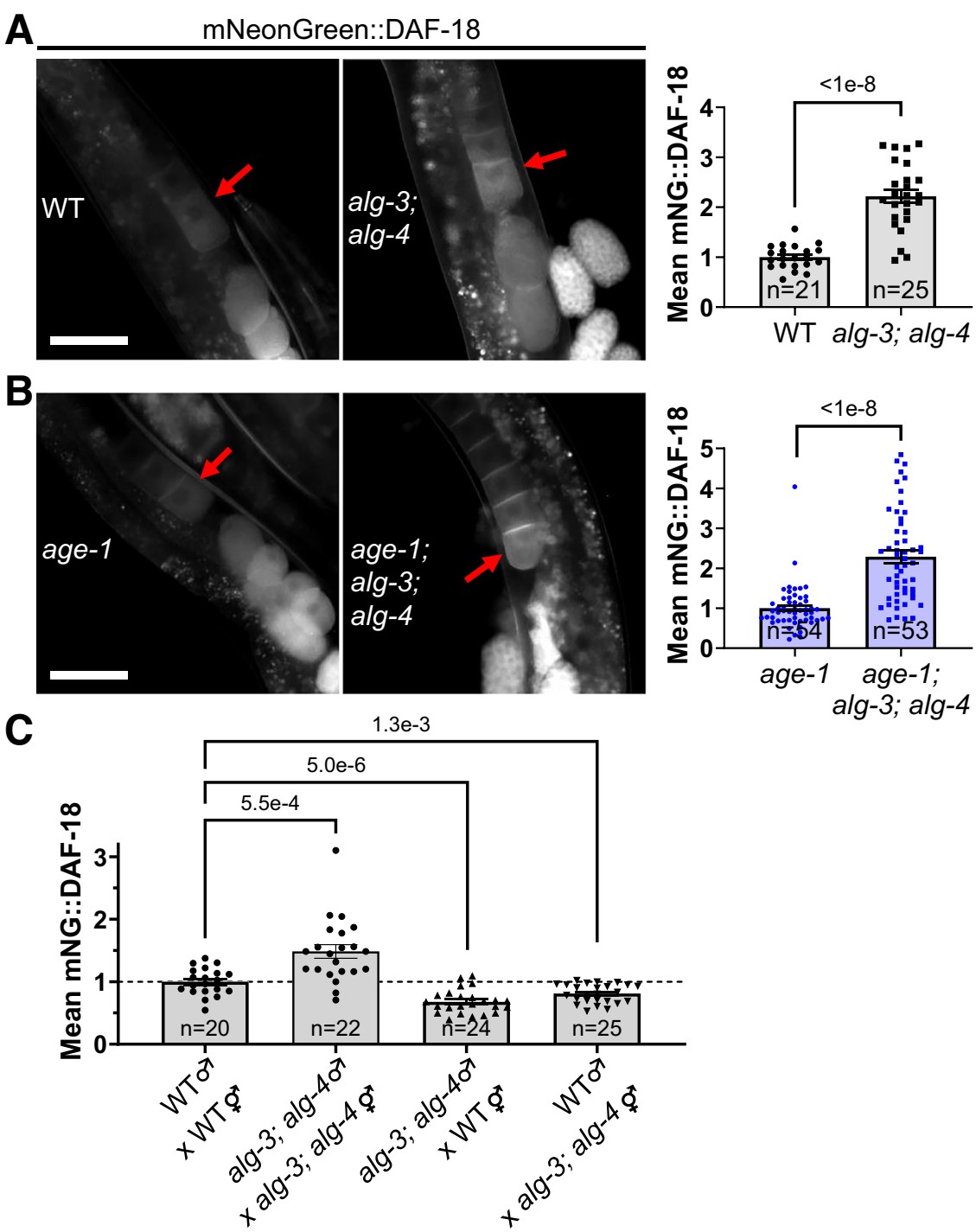

**Figure 7. ALG-3/4 inhibits DAF-18 expression.**

(**A, B**) Endogenously-tagged mNeonGreen::3xFLAG::DAF-18 (mNeonGreen::DAF-18 or mNG::DAF-18) in the −1 oocyte (red arrow) of adults is increased by the *alg-3(tm1155); alg-4(ok1041)* mutation in the WT (2.2-fold increase) (**A**) and *age-1(hx546)* (2.3-fold increase, *n* = 53–54) (**B**) backgrounds. Scale bars: 50 µm. (**C**) WT or *alg-3; alg-4* mutants were crossed to each other, and their progeny were imaged for mNG::DAF-18. All strains were homozygous for mNG::DAF-18. Unpaired two-tailed tests were used for comparisons: Mann-Whitney tests in (**A–C**) and t-tests in (**C**). Sample sizes for (**A–C**) are indicated in the graphs and represent the number of biological replicates (animals). Error bars are s.e.m. for (**A–C**). Source data are available online for this figure.

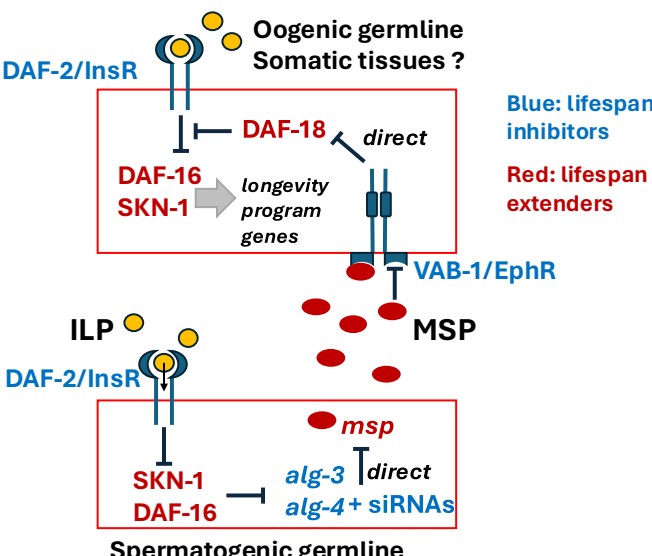

**Figure 8. Model of the interactions between IIS and endogenous RNAi.**

Summary model of the observed interactions between the IIS and the *alg-3/4* endogenous RNAi pathways. In the spermatogenic germline (bottom), Insulin-like peptides (ILP) activate DAF-2 receptor. This results in restriction of DAF-16 and SKN-1 nuclear localization and de-repression of *alg-3* and *alg-4* genes. ALG-3/4 bound to RDE-4-dependent siRNAs initiate the RNAi silencing pathway, repressing genes coding for secreted MSP proteins. When ALG-3/4 is disrupted, it no longer inhibits MSP expression, likely resulting in elevated MSP secretion, inhibition of VAB-1/EphR signaling, and elevated DAF-18/PTEN protein levels in tissues where DAF-16 and SKN-1 are critical for regulation of longevity program genes. This DAF-18 elevation in *alg-3/4(-)* is thought to ultimately contribute to enhanced suppression of IIS in a weak partial loss-of-function *age-1* mutant and lifespan extension.

et al, 2019) fuel images of cross-tissue and cross-generational regulatory networks of transported small RNAs. The ideas about the importance of heritable and systemic RNAi processes in adaptation to the environment and in evolution are also popular (Harris et al, 2025). Our findings demonstrate that small RNAs do not need to move beyond cells of origin to have systemic effects and to modulate key signaling pathways, such as IIS and Ephrin. At the same time, as we show Argonaute abundance regulation through IIS, it becomes clear that small RNA-based regulation of endogenous genes can be modulated by multiple inputs into IIS, such as nutrient abundance and environmental stress. In *C. elegans*, the germline is the site of numerous siRNA-dependent processes key to germline development and reproductive success. Therefore, the coordination of IIS activation via nutrients with germ cell production can be modulated by IIS-dependent ALG-3/4 upregulation. It would be interesting to identify insulin-like peptides (ILPs) promoting ALG-3/4 expression in spermatogenic germline and the tissue(s) they originate from (Fig. 8). Also, testing whether human IIS activation modulates Argonaute protein levels would be important.

Why does ALG-3/4 have a milder effect on lifespan in the WT background? The phosphorylation of PIP$_2$ by AGE-1 is likely not sufficiently opposed by the increase in DAF-18 caused by the *alg-3; alg-4* mutations to produce a strong decrease in IIS and extension of lifespan. However, in *age-1; alg-3; alg-4* mutants, the combination of decreased AGE-1 function and increased DAF-18 levels may

allow sufficient PIP$_3$ dephosphorylation (Figs. 8 and EV7). Moreover, our findings do not exclude the possibility that ALG-3/4 deficiency opposes IIS through additional mechanisms not involving DAF-18.

We bring attention to the physiological roles of secreted MSP domain proteins, as our results and earlier studies are consistent with the possibility that they mediate the non-cell-autonomous effects of *alg-3/4* loss. The MSP domains of the *Drosophila* and mammalian VAPB proteins were shown to be cleaved and secreted (Tsuda et al, 2008). Importantly, the MSP domains of VAP proteins, including *C. elegans* VPR-1, have ~25% identity with sperm MSPs and can induce oocyte maturation when injected into gonads lacking sperm, presumably through EphR inhibition (Tsuda et al, 2008). VPR-1 is expressed in somatic tissues, including neurons, and neuronally-expressed VPR-1 affects muscle morphology and fat accumulation in the muscle in a non-autonomous manner (Han et al, 2013, 2012). Interestingly, neuronal VPR-1 cooperates with muscle DAF-16 in regulating fat metabolism (Han et al, 2013). The VPR-1 connection to DAF-16 is thought to occur through its secreted MSP domain and resulting VAB-1/EphR and DAF-18/PTEN changes (Han et al, 2013). Here, we propose a similar function for germline MSPs.

Both DAF-18 overexpression and a *vab-1* mutation led to lifespan extension by 30% and 15%, respectively (Brisbin et al, 2009). For MSP-induced DAF-18 elevation and resulting activation of DAF-16 to influence lifespan, it should take place in tissues controlling lifespan extension. Importantly, gustatory ASI neurons inhibit lifespan via the IIS pathway (Alcedo and Kenyon, 2004), and VAB-1 is most highly expressed in ASI (Grossman et al, 2013). Therefore, there is an intriguing possibility that documented sensory control of *C. elegans* lifespan (Alcedo and Kenyon, 2004; Apfeld and Kenyon, 1999) is executed, at least in part, through MSP/VAB-1/DAF-18 connection. We report the effect of *alg-3/4* mutations on oocyte mNeonGreen::DAF-18 levels. This transgenic system does not allow us to estimate the possible effects of mutants on neuronal DAF-18 adequately. Given that long-lived *vab-1* mutants showed increased DAF-18 levels in neurons, in addition to oocytes (evaluated by antibody staining) (Brisbin et al, 2009), the increased secretion of MSPs in *alg-3; alg-4* mutants may result in elevated neuronal DAF-18 levels, which were shown to be sufficient to extend lifespan (Masse et al, 2005).

Although our results support a model where secreted MSPs affect lifespan through DAF-18 and DAF-16 in somatic tissues, we cannot exclude the possibility that other direct ALG-3/4 targets contribute to lifespan control. For example, the reduced level of cytoskeletal MSPs seen in *rde-4* and *alg-3/4* mutants may affect (enhance or reduce) the secretion of other signaling molecules enriched in the spermatogenic germline, such as INS-37, a known DAF-2 antagonist (Zheng et al, 2018). An important component of extended lifespan in *C. elegans* is the propagation of DAF-16/FOXO activation between tissues. This is achieved, in part, by FOXO-dependent regulation of DAF-2 agonist or antagonist insulins in the signaling cells (Murphy et al, 2007). The spermatogenic germline of *C. elegans* may serve both as an insulin signal acceptor tissue and a signaling tissue secreting MSPs and ILPs, and play a role in inter-tissue communication relevant to aging control. In higher organisms, the interplay between insulin and secreted MSP signaling may take place in the nervous system.

Overall, our work highlights the role of endogenous siRNAs in aging and the non-autonomous reach of RNA-based gene regulation. In our investigation, we discovered novel genetic interactions between conserved components of RNAi, IIS, FOXO, and Eph signaling, which may be informative in future developments of cancer or aging therapies.

## Methods

### Reagents and tools table

| Reagent/Resource | Reference or Source | Identifier or Catalog Number |
| --- | --- | --- |
| **Experimental models** | | |
| *C. elegans* wild type | *Caenorhabditis* Genetics Center | N2 |
| age-1(hx546) II | *Caenorhabditis* Genetics Center | TJ1052 |
| rde-1(ne219) V | *Caenorhabditis* Genetics Center | WM27 |
| rde-4(ne301) III | *Caenorhabditis* Genetics Center | WM49 |
| age-1(hx546) II; rde-4(ne301) III | Grishok Lab | AGK839 |
| age-1(hx546) II; rde-1(ne219) V | Grishok Lab | AGK841 |
| rde-4(ne301) III (6x outcrossed, henceforth "6x") | This study | AGK873 |
| age-1(hx546) II (6x) | This study | AGK875 |
| age-1(hx546) II; rde-4(ne301) III (6x) | This study | AGK886 |
| alg-4(ok1041) III; alg-3(tm1155) IV (6x) | This study | AGK950 |
| age-1(hx546) II; alg-4(ok1041) III; alg-3(tm1155) IV (6x) | This study | AGK951 |
| rde-1(ne219) V (6x) | This study | AGK870 |
| age-1(hx546) II; rde-1(ne219) V (6x) | This study | AGK882 |
| age-1(hx546) II; nrde-3(gg66) X (6x) | This study | AGK998 |
| daf-16(mgDf50) I; age-1(hx546) II | This study | AGK843 |
| daf-16(mgDf50) I; age-1(hx546) II; rde-4(ne301) III | This study | AGK858 |
| daf-16(mgDf50) I | *Caenorhabditis* Genetics Center | GR1307 |
| daf-2(e1370) III (5x outcrossed) | This study | AGK948 |
| daf-2(e1370) III rde-4(ne301) III (5x outcrossed) | This study | AGK949 |
| alg-3(tor141[GFP::3xFLAG::alg-3]) IV | *Caenorhabditis* Genetics Center | JMC205 |
| daf-2(e1370) III; alg-3(tor141) IV | This study | AGK985 |
| daf-16(mgDf50) I; alg-3(tor141) IV | This study | AGK987 |
| daf-16(mgDf50) I; daf-2(e1370) III; alg-3(tor141) IV | This study | AGK986 |
| age-1(hx546) II; alg-3(tor141) IV | This study | AGK947 |
| daf-16(hq23[daf-16::GFP]) I | *Caenorhabditis* Genetics Center | MQD1543 |
| daf-16(hq23) I; rde-4(ne301) III | This study | AGK968 |

| Reagent/Resource | Reference or Source | Identifier or Catalog Number |
| --- | --- | --- |
| daf-16(hq23) I; age-1(hx546) II | This study | AGK970 |
| daf-16(hq23) I; age-1(hx546) II; rde-4(ne301) III | This study | AGK972 |
| tnIs13[pie-1p::vab-1::GFP + unc-119(+)] ltIs44[pie-1p::mCherry::PH(PLC1delta1) + unc-119(+)] V | *Caenorhabditis* Genetics Center | DG2160 |
| alg-4(ok1041) III; alg-3(tm1155) IV; tnIs13 ltIs44 V | This study | AGK1026 |
| daf-18(utx19[mNG::3xFlag::daf-18]) IV (1x) | This study | AGK1021 |
| alg-4(ok1041) III; alg-3(tm1155) daf-18(utx19) IV | This study | AGK1023 |
| age-1(hx546) II; daf-18(utx19) IV | This study | AGK1022 |
| age-1(hx546) II; alg-4(ok1041) III; alg-3(tm1155) daf-18(utx19) IV | This study | AGK1024 |
| *E. coli* OP50 | *Caenorhabditis* Genetics Center | OP50 |
| **Oligonucleotides and other sequence-based reagents** | | |
| AGTGCGACATTGATATCCGT | This study | act-3 qPCR F |
| TCTTGATCTTCATGGTTGATGG | This study | act-3 qPCR R |
| ACAAGCGTCACCACTCTCATCC | This study | alg-3 qPCR F |
| TGTTGAGACAGGCCAATTCCGA | This study | alg-3 qPCR R |
| GCCATCAGTTGCAGCTATCGT | This study | alg-4 qPCR F |
| AGACAATGATGCGAGCTGGCT | This study | alg-4 qPCR R |
| CTTGTCGTTTATCCCAGATGGCT | This study | dct-9 qPCR F |
| CCGGGCTTCAGGATTCCACA | This study | dct-9 qPCR R |
| AGCTTGTTAACGGAGGTGCC | This study | ssp-16 qPCR F |
| TCACGACGTCCTTGGATCCA | This study | ssp-16 qPCR R |
| ACGGAGAGTTTGAAGAAGGACGA | This study | melo-3 qPCR F |
| GGAGTTTGGCTTCCTCTGGATCA | This study | melo-3 qPCR R |
| **Chemicals, Enzymes and other reagents** | | |
| miRNeasy Mini Kit | Qiagen | 217004 |
| Direct-zol RNA Miniprep | Zymo Research | R2051 |
| Maxima Reverse Transcriptase | Thermo Scientific | EP0741 |
| Luna Universal qPCR Master Mix | New England Biolabs | M3003L |
| **Software** | | |
| Wormbase | https://wormbase.org | |
| GraphPad Prism 10 | https://graphpad.com | |
| R | https://r-project.org | |
| BioRender | https://biorender.com | |
| **Other** | | |
| Illumina NovaSeq 6000 | Illumina | |

## C. elegans maintenance

Worms were cultured and assayed as hermaphrodites at 20 °C (unless indicated otherwise) on solid nematode growth media (NGM) seeded with *E. coli* OP50. For all assays, worms used were always grown for at least 3 generations after a starvation event (including thawing) to prevent known transgenerational effects (Jobson et al, 2015; Webster et al, 2018; Rechavi et al, 2014).

## Lifespan and healthspan assays

The experimenter was blinded to each condition (e.g., strain). For hermaphrodites, 20–30 young (day 1) adults were transferred to each NGM plate and were moved to a new plate every 2 days until egg-laying ceased. For males, 4 L4s were transferred to each plate, and the number of worms per plate was kept strictly constant to 4 (or less only when too few animals were alive) to account for the negative effects on lifespan from the presence of other males (Gems and Riddle, 2000; Shi et al, 2017). Note that over half of the male population typically escaped from plates over the duration of their lifespan. Worms were transferred to a fresh plate every week. Drugs affecting progeny production, such as Fluorodeoxyuridine (FUdR) were not used. Animals were scored every 2–3 days and were considered dead when they stopped exhibiting spontaneous movement and failed to move in response to (1) a gentle touch of the tail, (2) a gentle touch of the head, and (3) gently lifting the head. Animals dying from unnatural causes (internal hatching of embryos, bursting, intestine leaking, or crawling off the plate) were right-censored and omitted from the survival curve trace but included in the lifespan's statistical analysis and reported in Dataset EV1. The log-rank test was used to assess significance in OASIS 2 (Han et al, 2016).

For the measurement of healthspan, each worm was first observed (without prodding) for 5 seconds for any spontaneous movement. Scoring for death was performed after all worms on a plate were scored for healthspan. Data from several plates were pooled. Statistics were performed with Fisher's exact test based on a 2×2 contingency table for mutant vs. control and number of worms with vs. without spontaneous movement.

## RNA extraction, RNA-sequencing, and qPCR

Three independent biological replicates of worm populations were used for RNA-sequencing, and an additional three for qPCR. Recently starved L1 worms were chunked on NGM plates seeded with concentrated OP50 bacteria. For RNA-sequencing, NGM plates contained half the amount of agar replaced with an equal amount of agarose. NGM chunks were removed the following day. Once reaching egg-laying adults, worms were bleached with an alkaline hypochlorite solution (NaOH 0.25 M, sodium hypochlorite 4–5% (Sigma Aldrich)) for age synchronization of their progeny. Eggs were collected in M9 buffer and allowed to hatch and arrest as L1 overnight. L1 worms were transferred to NGM (or NGM agar/agarose plates for RNA-sequencing) and grown until reaching the desired stage. For RNA-sequencing, this was the young adult stage (first day of adulthood, before internal egg development). Worms were collected and washed in M9 and added to Trizol Reagent (Invitrogen). Worms in Trizol were lysed by freeze-thawing from −80 °C to 4 °C three times. Total RNA was purified from the lysate using the miRNeasy kit (Qiagen),

including the optional RWT wash step, for RNA-sequencing or the Direct-zol kit (Zymo Research), including DNAse I treatment, for qPCR, following manufacturer instructions. RNA was sent to Novogene Corporation for library preparations and next-generation sequencing of mRNA, lncRNA, and small RNA.

For mRNA- and lncRNA-sequencing, ribosomal RNA was depleted, cDNA fragments of 250–300 bp were enriched, and sequencing was performed with Illumina NovaSeq 6000 with 150 bp paired-end reads. For small RNA-sequencing, 5' monophosphate small RNAs were enriched, and sequencing was performed with Illumina NovaSeq 6000 with single-end 50 bp reads.

For qPCR, WT and mutant *daf-2(e1370)*, *daf-16(mgDf50)*, and *daf-2; daf-16* strains were in the GFP::ALG-3 background. 100 ng of RNA was used for cDNA synthesis using Maxima Reverse Transcriptase (Thermo Scientific), random hexamer primer, and following the manufacturer's standard procedures. Primers were designed to span exon-exon junctions and did not amplify in samples without RT. The *act-3* gene was used as a normalization control. Luna® Universal qPCR Master Mix (New England BioLabs) was used for qPCR in a QuantStudio 6 Pro (Applied Biosystems) according to manufacturer specifications. For comparisons, cycle threshold (Ct) values of a given gene in a mutant were first normalized to *act-3* and to the *act-3*-normalized WT young L4 gene Ct of the same biological replicate (ΔΔCt). Comparisons between genes in young L4s used the WT late L4 as the normalization calibrator instead.

## Bioinformatic analyses

Bioinformatic analysis of sequencing data was performed as in previous work (Liontis et al, 2023). Briefly, HISAT2 (Kim et al, 2015) was used to map mRNA reads. StringTie was used for quantification of reads (Pertea et al, 2015), FPKM normalization was performed, and edgeR was used for differential expression (Robinson et al, 2010). Analysis of differentially expressed siRNAs from the small RNA-sequencing data was performed with a custom script using QuasR (Gaidatzis et al, 2015). Briefly, small RNAs were mapped to the *C. elegans* genome (WBcel235, bioproject PRJNA13758, release WS283, NCBI Refseq accession: GCF_000002985.6, an annotation that lacks piRNAs, which were not considered for siRNA analysis). Alignment was done using Bowtie (Langmead et al, 2009) with parameter maxHits = 50. Counting was done using parameter orientation = "same" for sense siRNAs and "opposite" for antisense siRNAs. DESEQ2 (Love et al, 2014) was used for the analysis of differential expression. Differentially expressed genes considered significant were those with $p < 0.05$, to maximize the detection of genes that were primarily used for subsequent overlaps with external datasets.

To perform gene set overlaps for enrichment, a common name for each gene, "Public Name", was extracted from all datasets by inputting gene names (in whichever available format) to WormBase's Gene Name Sanitizer, selecting the Suggested Match WBGene ID, and inputting it to WormBase's SimpleMine tool, which outputted the "Public Name". To properly correspond and overlap our sequencing data to the Argonaute mutant small RNA dataset by (Seroussi et al, 2023) piRNAs and miRNAs were first filtered out from the latter dataset.

The representation factor (RF) determines whether the number of genes overlapping among two sets is higher than expected given

their relative size in the genome. RF = $(n_{1,2})/[(n_1 \times n_2)/N]$, with $n_{1,2}$ number of genes common to sets 1 and 2, $n_1$ number of genes in set 1, $n_2$ number of genes in set 2, $N$ total number of genes considered ($N = 20,000$ genes in the *C. elegans* genome).

The two-sided Fisher's exact test was used to measure the significance of gene overlap enrichment. For screening among all Argonaute mutant small RNA datasets (Seroussi et al, 2023) in Dataset EV6, the one-sided test was used because under-representation was less informative.

For ALG-3/4 "positive" and "negative" targets, Supplementary Table S1 from (Conine et al, 2013) was used, which includes genes for which corresponding 26 G RNA and mRNA levels are dependent on *alg-3; alg-4* at 25 °C. For RDE-1 targets and datasets of small RNAs dependent on each known Argonaute, the dataset of 26 G small RNA-sequencing in Argonaute mutants from (Seroussi et al, 2023) was used. To obtain the list of genes with an MSP domain in *C. elegans*, we queried WormMine (WS294) for the MSP domain protein motif (PFAM PF00635) and converted protein IDs to unique gene IDs. The resulting list of genes was overlapped with the differentially expressed genes in *age-1; rde-4* vs *age-1* mutants from our dataset.

### Brood size

The brood size assay was performed as done previously (Machiela et al, 2020). Briefly, one L4 worm was placed on each NGM plate and was moved to a new plate daily during progeny production. The number of progeny reaching at least the L2 stage was counted. For brood size assays involving crossing, L3 males and hermaphrodites (P0) were shifted from 20 °C to 25 °C, and their progeny (F1) were counted. Successful mating was determined based on a significant (near 50%) presence of male F1.

### Fluorescence microscopy

Imaging was performed using the Zeiss AxioImager Z1. Worms were mounted on 2% agarose pads, immobilized with 10–20 mM levamisole in M9 buffer, placed under a thin glass coverslip, and immediately imaged. A constant exposure was used for images shown in the same figure panel. Image analysis utilized the ImageJ (Fiji) software (Schindelin et al, 2012). Blinded experimenters measured the mean fluorescence of manually delimited areas of interest. Controls (e.g., WT) were always present on the exact same surface (same slide, agarose pad, and coverslip) as experimental animals (e.g., mutants) subjected to the same exposure. The experimental signal was normalized to controls in each replicate, and results were pooled for the graphs presented. Random counterbalancing of the order in which strains were imaged was done between each independent replicate. Staging of worms for GFP::ALG-3 imaging and qPCRs utilized previously defined L4 substages (Mok et al, 2015), with L4.1–L4.3 as young L4, L4.4–L4.6 as moderate L4, and L4.7–L4.9 as late L4. Moderate Day 1 adults were a few hours after the young adult stage, around the appearance of the first embryo. Day 2 adults are 24 h after the young adult stage. VAB-1::GFP was assessed in the proximal-1 oocyte in animals 24 h after the L4–young adult stage, similarly to what has been done previously (Cheng et al, 2008; Harris et al, 2006). An experimenter blinded to the genotype determined whether VAB-1::GFP vesicles were small or large and whether

localization was "perimembrane", i.e., enrichment of vesicles surrounding the cell membrane in an orderly manner, "random", i.e., no enrichment of vesicles for the membrane area, where vesicles had no clear pattern, or "intermediate", where vesicles incompletely surrounded the cell membrane and multiple interior vesicles were present (all phenotypes shown in Fig. EV6). Enrichment of VAB-1 vesicles near the nucleus occurred in all cases. Confocal microscopy images were acquired with a Nikon A1R confocal microscope, using Nikon Elements software driving the galvanometer scanner. mNeon-Green::DAF-18 was measured in the cytoplasm of the proximal $-1$ oocyte of worms 24 h after the young adult stage, similarly to previous research (Brisbin et al, 2009).

### Parental effects on DAF-18

All strains used were homozygous for mNeonGreen::DAF-18. Four males and one hermaphrodite at the L3–L4 stage were transferred to each plate, with several plates for each cross. Hermaphrodites were considered to have mated when a significant number of progeny (F1) were males (up to 50%). F1 late L4 hermaphrodites identified as cross-progeny (not older than sibling F1 males) were transferred to a new plate to prevent further crossing. Animals were imaged for mNeonGreen::DAF-18 24 h after the young adult stage. Three independent replicates were performed.

### Statistical analyses and graphs

For comparisons of means, the F-test for variance and Anderson-Darling, D'Agostino-Pearson, Shapiro-Wilk, or Kolmogorov-Smirnov tests for normality were first run. The resulting appropriate unpaired Student's t-test (equal vs. unequal variance) or Mann-Whitney test (nonparametric) was then used to assess a two-tailed significant difference between two groups. Fisher's exact test was used for discrete data expressed as a proportion of the population (graphed values bounded between 0 and 1). The $n$ number represents the number of animals. Tests were performed with GraphPad Prism 10.

Venn diagrams were generated with the R package eulerr by Johann Larsson using the parameter input = "union". For improved readability of graphs, colored labels indicate the value corresponding to an entire circle of the matching color, whereas black labels indicate a delimited intersecting/overlap region. Lifespan, health-span, and bar graphs (error bars represent the standard error of the mean) were generated with GraphPad Prism 10. Heat maps and scatter plots were constructed in R. The graphical abstract was created in BioRender [Liontis, T. (2025) https://BioRender.com/jjgvwuy]. In cases where $p$ values are indicated as $p < x$, the x represents the smallest value that our software could compute for a given statistical test.

## Data availability

Worm strains generated in this study are available upon request or through the Caenorhabditis Genetics Center (CGC) (https://cgc.umn.edu/). Raw and processed RNA-sequencing data have been deposited in the NCBI Gene Expression Omnibus (GEO) (Barrett et al, 2013) as GEO: GSE293149 and GEO: GSE293150 and are publicly available as of the date of publication.

The source data of this paper are collected in the following database record: biostudies:S-SCDT-10_1038-S44319-025-00682-4.

## Peer review information

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

## Acknowledgements

We thank Michael Blower for the generous help with confocal imaging, Gian Paolo Sepulveda and Karol Nawalaniec for insightful discussions and troubleshooting. This work was supported by a National Institutes of Health grants [R01 GM135199] to AG and [P40 OD010440] to the Caenorhabditis Genetics Center, and a Hevolution Foundation grant [HF-AGE-23-1268260-52] to AG.

## Author contributions

**Thomas Liontis**: Conceptualization; Data curation; Formal analysis; Investigation; Visualization; Methodology; Writing—original draft; Writing—review and editing. **Valentina T Pannarale**: Formal analysis; Investigation. **Andrés R Mansisidor**: Conceptualization; Investigation; Methodology. **Sasiru K Pathiranage**: Formal analysis; Investigation. **Jeeya Y Patel**: Data curation. **Alla Grishok**: Conceptualization; Supervision; Funding acquisition; Visualization; Writing—original draft; Project administration; Writing—review and editing

Source data underlying figure panels in this paper may have individual authorship assigned. Where available, figure panel/source data authorship is listed in the following database record: biostudies:S-SCDT-10_1038-S44319-025-00682-4.

## Disclosure and competing interests statement

The authors declare no competing interests.

# Expanded View Figures

**Figure EV1. Effects of interactions between *age-1(hx546)* and RNAi pathways on gene expression and lifespan.** ▶

**(A–C)** Genes upregulated in *age-1(hx546)* are significantly enriched for genes upregulated in *rde-4(ne301)* (h: RF = 4.5, $p < 1 \times 10^{-230}$) **(A)**, RDE-1 targets (Seroussi et al, 2023) (i: RF = 2.3, $p = 7.1 \times 10^{-175}$) **(B)**, and ALG-3/4 positive (j: RF = 3.5, $p = 6.6 \times 10^{-46}$) and negative (k: RF = 3.9, $p = 1.8 \times 10^{-63}$) targets (Conine et al, 2013) **(C)**. **(D)** Survival curves showing that the *rde-1(ne219)* mutation mildly and similarly extends lifespan in both WT (+ 5% MLS, n = 249 and 251 animals) and *age-1(hx546)* (+7.5% additional MLS, n = 145 and 152 animals) backgrounds. **(E)** Survival curves showing that the *alg-3(tm1155); alg-4(ok1041)* mutations do not extend lifespan in the WT (n = 177 and 183 animals) nor *age-1(hx546)* (n = 191 and 195 animals) backgrounds of male nematodes. **(F)** Survival curves showing that *age-1(hx546); nrde-3(gg66)* mutants live shorter than *age-1(hx546)* mutants (n = 65 and 72 animals). **(G)** Qualitative relative assessment of the number of worms moving spontaneously among those alive (relative spontaneous movement) in *age-1(hx546)* and *age-1(hx546); alg-3(tm1155); alg-4(ok1041)* strains. The lifespan assay for all technical replicates (technical rep) of each strain was started at the same time (N = 1 independent experiment). Each technical rep represents a different plate containing a population of worms. The experimenter was blinded to the identity of a strain for each technical replicate. Relative spontaneous movement score (assessed for each technical rep) of +++ means more, whereas ++ or + means relatively fewer worms exhibiting spontaneous movement; - means no spontaneous movement despite being alive. Color associated with each spontaneous movement score is assigned to a cell based on 1) the majority assessment among technical reps and 2) in case of a tie, continuity with the previous or next timepoint's score (e.g., *age-1; alg-3; alg-4* Day 39 is orange because its majority score is tied for + and - but the next timepoint is +). Not all technical replicates were necessarily assessed on all days. Gene set enrichment in **(A–C)** assessed by Fisher's exact test. Survival curves **(D–F)** include pooled data from all independent experimental replicates and are compared by the log-rank test. All percentage MLS changes are relative to WT. ns: not significant. *age-1(-): age-1(hx546)*. *rde-4(-): rde-4(ne301)*.

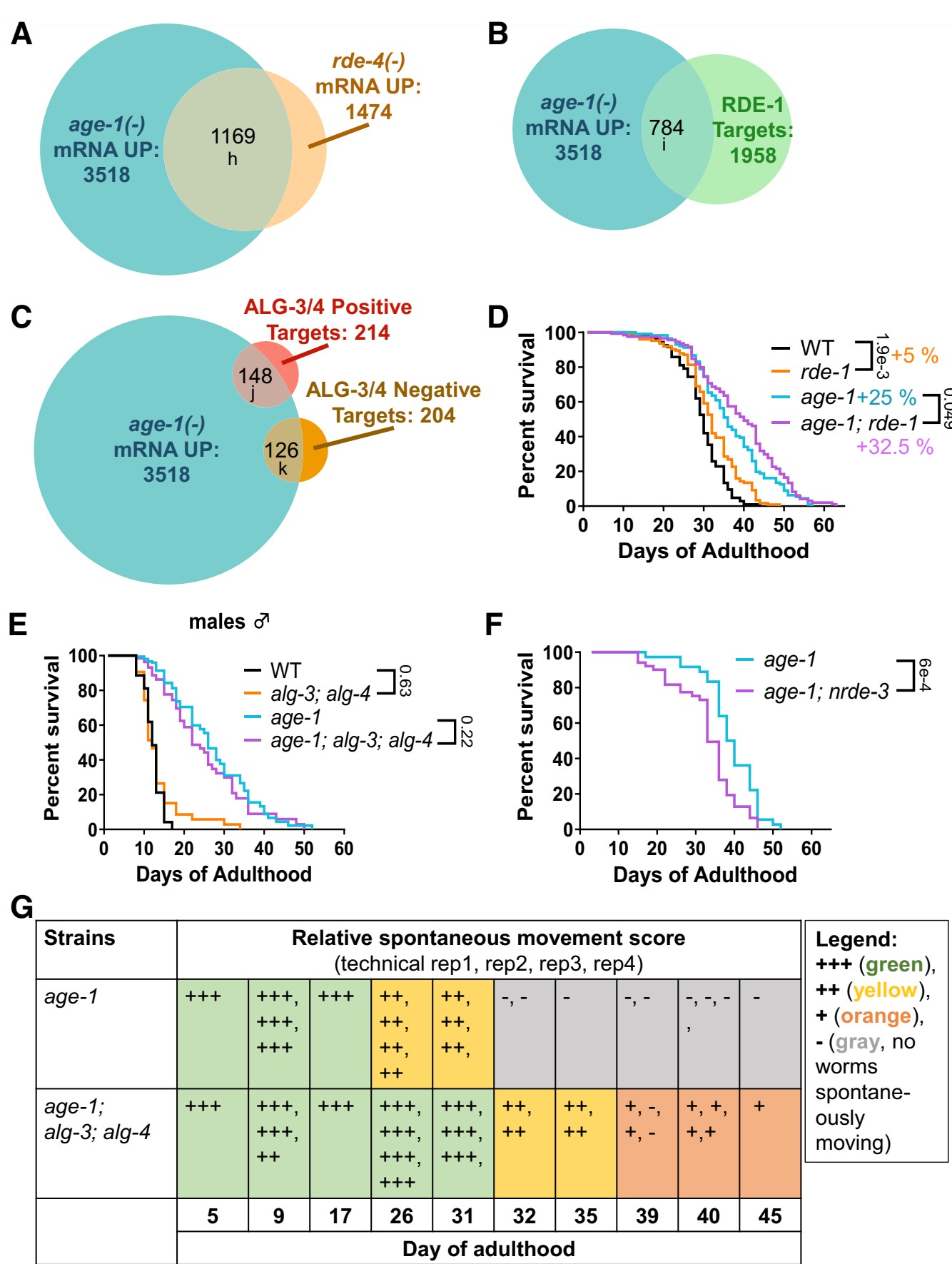

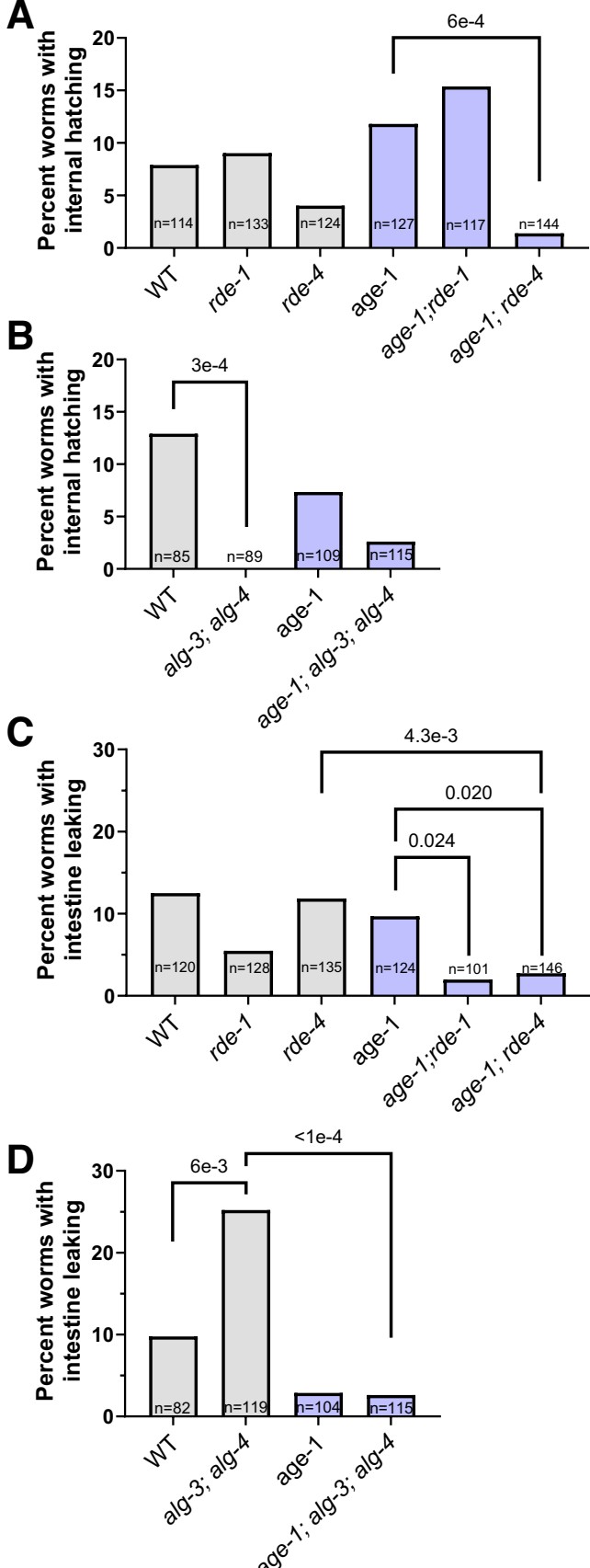

**Figure EV2. Unnatural deaths in RNAi and *age-1* mutants.**

(A, B) In WT and *age-1(hx546)* backgrounds, the proportion of *rde-1(ne219)*, *rde-4(ne301)* (A), or *alg-3(tm1155); alg-4(ok1041)* (B) mutant worms that died of internal hatching throughout their lifespan. (C, D) The proportion of worms that died from the intestine leaking out throughout the lifespan of the same strains. Sample sizes for (A–D) are indicated at the bottom of graphs and always represent the number of animals in all figures. Bars in (A–D) are compared using Fisher's exact test (with vs. without phenotype, in strain A vs. strain B) in data pooled from multiple independent replicates.

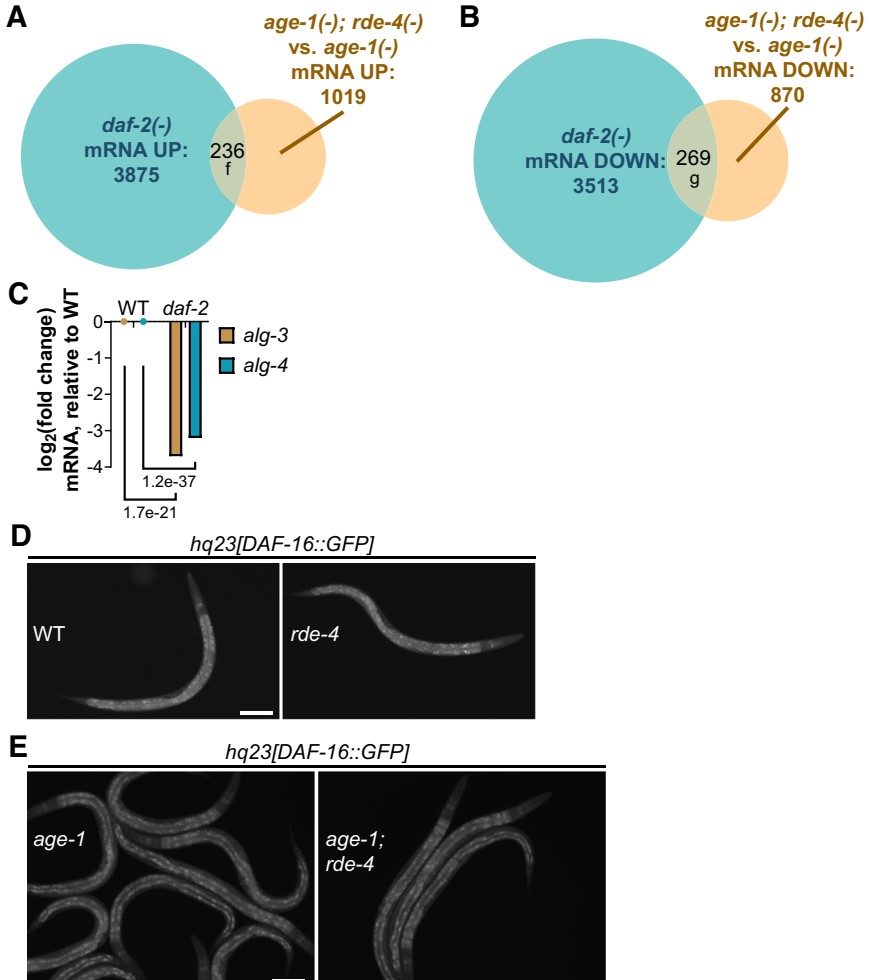

**Figure EV3.  Disruption of ALG-3/4 in strong IIS mutant _daf-2(e1370)_.**

(A, B) Significant overlaps between genes upregulated in _daf-2(e1370)_ compared to WT (Zullo et al, 2019) and in _age-1(hx546); rde-4(ne301)_ compared to _age-1(hx546)_ (f: RF = 1.2, $p = 8.3 \times 10^{-6}$) (A), as well as their downregulated genes (g: RF = 2.0, $p = 8.5 \times 10^{-42}$) (B). (C) Significant downregulation of _alg-3_ ($p = 1.2 \times 10^{-37}$) and _alg-4_ ($p = 1.7 \times 10^{-21}$) mRNA levels in _daf-2(e1370)_ according to published RNA-sequencing data ($n = 3$ independent worm populations) (Zullo et al, 2019). EdgeR was used for statistical analysis. (D, E) Representative images of endogenously-tagged DAF-16 in the _daf-16(hq23[daf-16::GFP])_ strain, where no obvious effect of the _rde-4(ne301)_ mutation is observed in WT (D) and _age-1(hx546)_ (E) backgrounds (L4 – young adult). Scale bars: 100 μm. Gene set enrichment in (A, B) was assessed by Fisher's exact test.

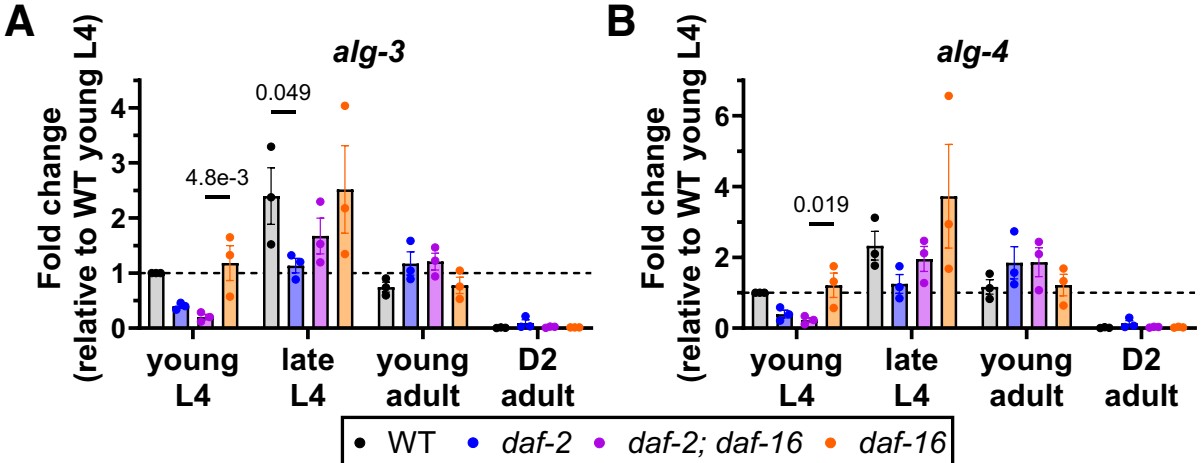

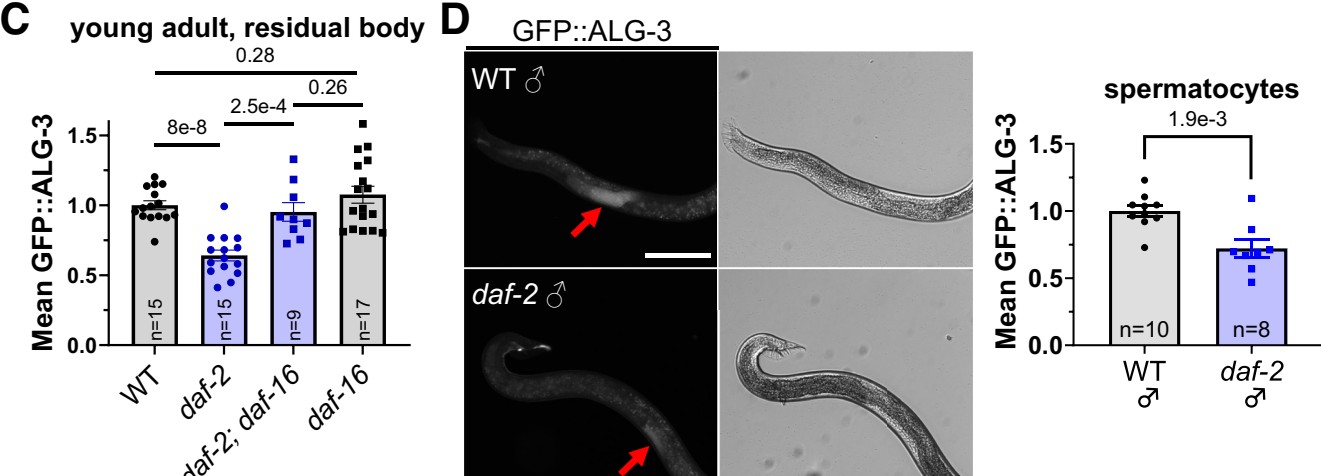

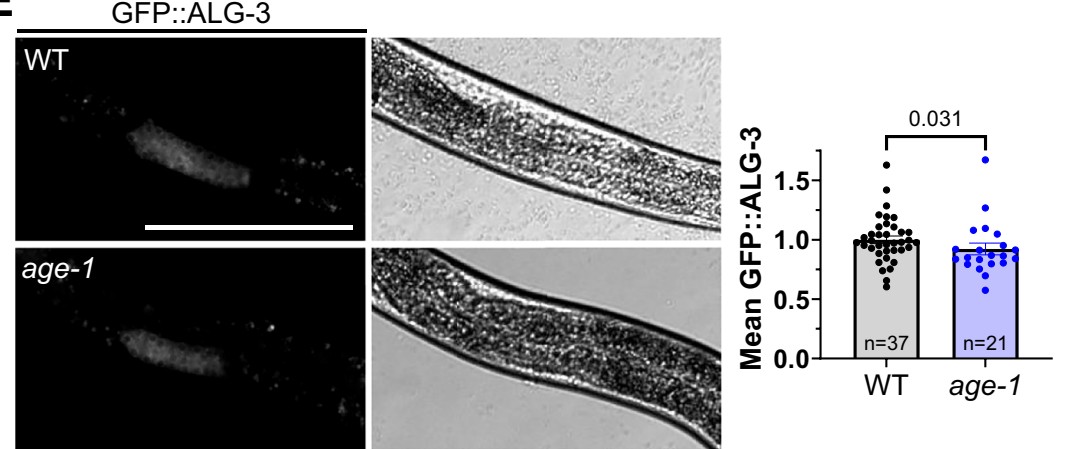

**Figure EV4.   DAF-2 promotes *alg-3* expression in a DAF-16-dependent manner.**

(A, B) Time-series qPCRs during young L4, late L4, early young adult, and D2 adult stages, detecting *alg-3* (A) and *alg-4* (B) mRNA expression (*n* = 3 independent worm populations). (C) Quantification of GFP::ALG-3 expressed in residual bodies of early young adults, shown in Fig. 4A. (D) Representative images of GFP::ALG-3 in spermatocytes (red arrow) of *daf-2(e1370)* and WT male young adults (left), quantified (right). (E) Representative images of GFP::ALG-3 in *age-1(hx546)* and WT L4 hermaphrodites (left), quantified in late L4 and young adults (right). Note the very small magnitude of change (−8%) in *age-1(hx546)*. Scale bar: 100 μm in (D, E). Sample sizes indicated in bar graphs of (C–E) represent the number of biological replicates (animals). Comparisons in (A–E) used unpaired two-tailed tests: t-tests in (A–D) and Mann-Whitney tests in (C, E). Error bars are s.e.m. for (A–E).

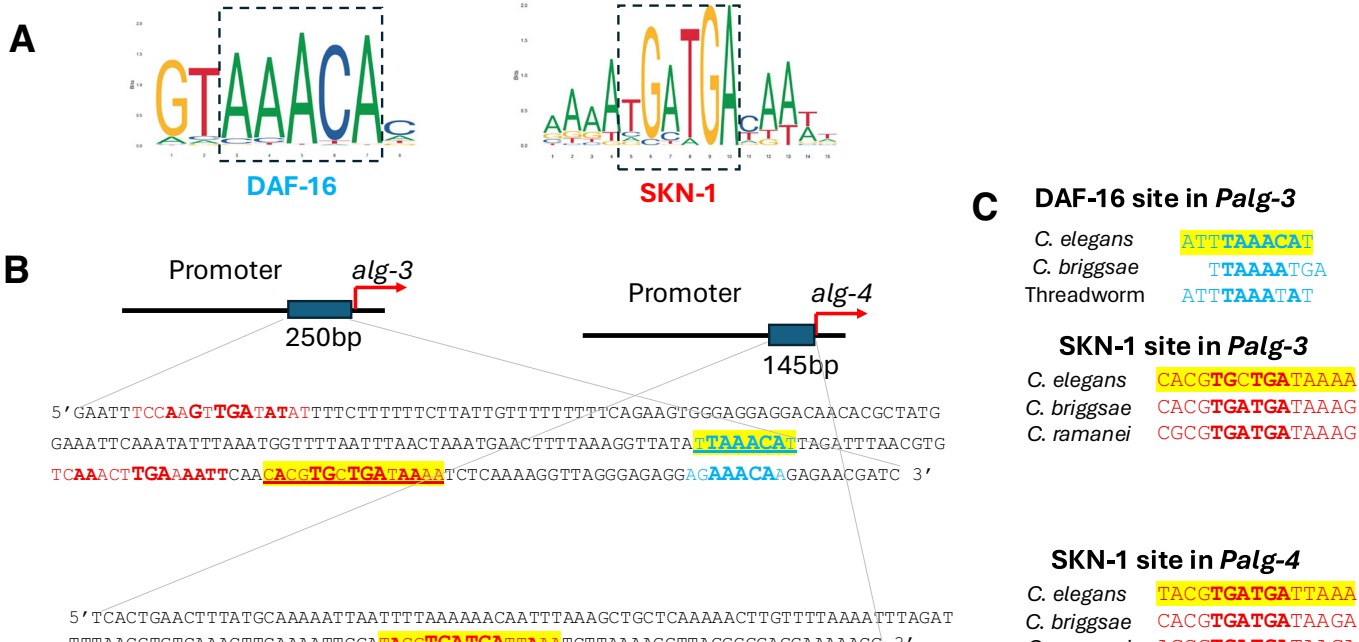

**Figure EV5. Identification of potential DAF-16 and SKN-1 binding sites at *alg-3* and *alg-4* promoter regions.**

(A) DAF-16 (Matrix ID: MA1446.2) and SKN-1 (Matrix ID: MA0547.2) ChIP-seq-based binding site consensus sequence logos from the 10th release (**2024**) JASPAR database. JASPAR is an open-access database of curated, non-redundant transcription factor (TF) binding profiles stored as position frequency matrices (PFMs) and TF flexible models (TFFMs) for TFs across multiple species in six taxonomic groups. (B) Potential DAF-16 and SKN-1 binding sites at the promoter sequences of *alg-3* and *alg-4* genes; sites showing conservation are marked in yellow. (C) Conservation of potential DAF-16 and SKN-1 binding sites in indicated nematode species (based on information from the UCSC Genome Browser on *C. elegans* Feb. 2013 (WBcel235/ce11)).

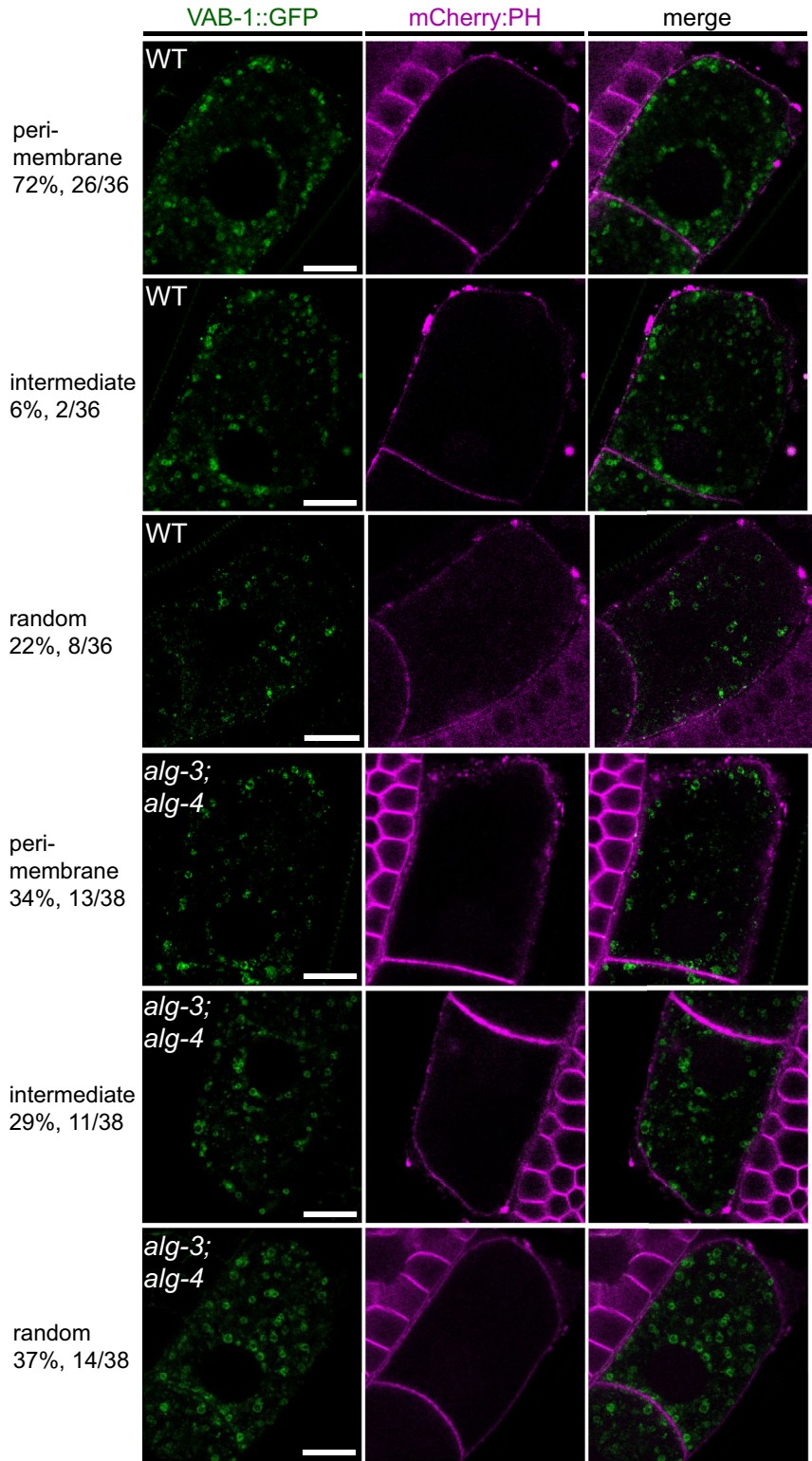

**Figure EV6.  VAB-1::GFP phenotypes in WT and *alg-3; alg-4* mutants.**

VAB-1::GFP and mCherry:PH confocal microscopy images of adult −1 oocytes with each VAB-1::GFP localization phenotype (perimembrane, intermediate, and random) in WT and *alg-3(tm1155); alg-4(ok1041)* mutants. Images for the perimembrane phenotype in WT and random phenotype in *alg-3(tm1155); alg-4(ok1041)* mutants are identical to those in Fig. 6B. (*Left*) The prevalence of each localization phenotype is shown in percentages, as well as the underlying (number of worms with the phenotype)/(total number of worms). These percentages are represented in Fig. 6C. Scale bars: 10 μm.

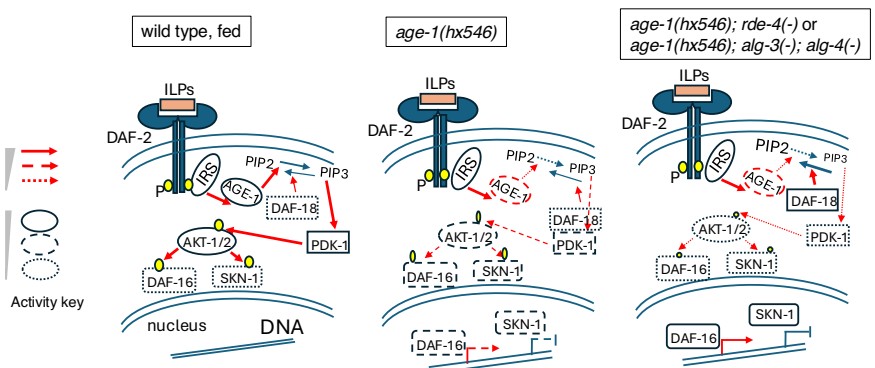

**Figure EV7.** Schematic representation of the proposed mechanism responsible for the synergistic lifespan extension seen in *age-1*; *alg-3/4* mutants.

