## [Peer Review File · EMBO Reports]

Mutual regulation of spermatogenesis-specific Argonautes and Insulin/IGF-1 signaling in aging

Alla Grishok, Thomas Lontis, Valentina Pannarale, Andrés Mansisidor, Sasiru Pathiranaage, and Jeeya Patel

Corresponding author(s): Alla Grishok (agrishok@bu.edu)

Review Timeline:

Submission Date:	29th Apr 25
Editorial Decision:	22nd May 25
Revision Received:	18th Sep 25
Editorial Decision:	22nd Oct 25
Appeal Received:	23rd Oct 25
Editorial Decision:	1st Dec 25
Revision Received:	1st Dec 25
Accepted:	11th Dec 25

Editor: *Esther Schnapp*

Transaction Report:

Dear Dr. Grishok,

Thank you for the submission of your manuscript to EMBO reports. We have now received the full set of referee reports that is pasted below.

As you will see, all referees acknowledge that the ms reports interesting findings. However, they also all have several suggestions for how the study should be strengthened and improved, including data presentation and manuscript writing. I think all suggestions are good and should be addressed, except may be for point 2 by referee 1, as I am not sure how difficult or time-consuming these CRISPR experiments are? Can you please let me know what you think about all referee comments and whether you will be able to address them? We can also discuss the revisions in a video chat, if you like.

If we can agree on a revision plan, I would like to invite you to revise your manuscript with the understanding that the referee concerns must be fully addressed and their suggestions taken on board. Please address all referee concerns in a complete point-by-point response. Acceptance of the manuscript will depend on a positive outcome of a second round of review. It is EMBO reports policy to allow a single round of major revision only and acceptance or rejection of the manuscript will therefore depend on the completeness of your responses included in the next, final version of the manuscript.

We realize that it is difficult to revise to a specific deadline. In the interest of protecting the conceptual advance provided by the work, we recommend a revision within 3 months (22nd Aug 2025). Please discuss the revision progress ahead of this time with the editor if you require more time to complete the revisions.

- 1) A data availability section providing access to data deposited in public databases is missing. If you have not deposited any data, please add a sentence to the data availability section that explains that.
- 2) Your manuscript contains statistics and error bars based on $n=2$. Please use scatter blots in these cases. No statistics should be calculated if $n=2$.

3) We replaced Supplementary Information with Expanded View (EV) Figures and Tables that are collapsible/expandable online. A maximum of 8 EV Figures can be typeset. EV Figures should be cited as 'Figure EV1, Figure EV2' etc... in the text and their respective legends should be included in the main text after the legends of regular figures.

5) a complete author checklist, which you can download from our author guidelines <https://www.embopress.org/page/journal/14693178/authorguide>. Please insert information in the checklist that is also reflected in the manuscript. The completed author checklist will also be part of the RPF.

6) Please note that all corresponding authors are required to supply an ORCID ID for their name upon submission of a revised manuscript (<https://orcid.org/>). Please find instructions on how to link your ORCID ID to your account in our manuscript tracking system in our Author guidelines <https://www.embopress.org/page/journal/14693178/authorguide#authorshipguidelines>

12) All Materials and Methods need to be described in the main text using our 'Structured Methods' format, which is required for all research articles. According to this format, the Methods section includes a Reagents and Tools Table (listing key reagents, experimental models, software and relevant equipment and including their sources and relevant identifiers) followed by a Methods and Protocols section describing the methods using a step-by-step protocol format. The aim is to facilitate adoption of the methodologies across labs. More information on how to adhere to this format as well as a downloadable template (.docx) for the Reagents and Tools Table can be found in our author guidelines: <https://www.embopress.org/page/journal/14693178/authorguide#structuredmethods>.

An example of a Method paper with Structured Methods can be found here: <https://www.embopress.org/doi/full/10.1038/s44320-024-00037-6#sec-4>

I look forward to seeing a revised form of your manuscript when it is ready.

Referee #1:

With this manuscript, Liontis and colleagues aim to determine the role of endogenous small interfering RNAs in animal aging. Using the nematode *C. elegans* as a model, the authors uncover the interaction between components of the endo-siRNA pathway and insulin/IGF-1 signaling and provide key insights into how ALG-3/4 Argonautes modulate the lifespan of *C. elegans*.

Overall, this study is interesting and sheds light on the role of endogenous siRNAs in controlling animal aging. As detailed below, substantial work will be needed to strengthen the conclusion and make this work suitable for publication:

1-As ALG-3 and ALG-4 are primarily linked to male-specific siRNAs, it seems essential to assess their effects on the aging of male animals as well. The phenotypic and molecular impacts may be even more significant and biologically relevant for those animals. Moreover, these experiments will also contribute to refining the model related to the cell non-autonomous role of ALG-3 and ALG-4 in this process.

2- Since the phenotypes observed in *hrde-1* and *nrde-3* mutants differ significantly from those of *alg-3* mutant animals, I am not convinced that they are the best genetic approach for studying the effect of ALG-3/4 on DAF-16 expression. The authors should instead use CRISPR-based genome editing to delete *alg-3* in the *zls356* strain to overcome the difficulty of genetic crossing and directly test the contribution of ALG-3/4 to DAF-16.

3- It appears that there are very few RDE-4-dependent small RNAs targeting the *lea-1* gene. Are they sufficient to affect its expression? The authors should monitor the expression of *Lea-1* mRNA to experimentally support its control by RDE-4.

4- Since the different mutants used in this study can affect germline integrity, it is important to include DIC images along with the fluorescent images found in Supplemental Figure S7 to determine whether the observed effect on LEA-1 expression is due to morphological differences.

5- It is quite challenging to determine any differences in VAB-1 localization in oocytes based on the images presented in Figure 6C. A set of images used to compile the data displayed in panels D and E should be included.

6- Even with some experience in the *C. elegans* small RNA field, I found this study rather complicated to follow. Careful editing will be necessary to make its key messages more accessible to readers.

Referee #2:

This paper by Liontis et al., explores roles for endogenous RNA interference (endo-RNAi) factors in regulating longevity in *C. elegans*. This is an important study because many pathways are known to influence lifespan in *C. elegans*, but few investigations have focused on the impact of endo-siRNA pathways in aging worms. While all tested loss of function mutants for endo-siRNA factors had minor, if any, lifespan phenotypes, they all seemed to enhance the extended lifespan of the insulin signaling mutant, *age-1(hx546)*, to various degrees (except *nrde-3*). RDE-4 participates in the generation of most endo-siRNAs, but not miRNAs, and small RNA sequencing showed depletion of primary endo-siRNAs used by the ALG-3/4, RDE-1 and ERGO-1 Argonautes for regulating target mRNAs. In addition to loss of *alg-3/4* enhancing the extended lifespan of *age-1* mutants, this strain also exhibited prolonged healthspan compared to loss of *age-1* alone. Yet, fertility was compromised more than loss of *age-1* or *alg-3/4* on their own. Their RNA-seq data pointed to DAF-16 activity as contributing to the enhanced lifespan resulting from loss of *rde-4* and *age-1* and, indeed, loss of *daf-16* in this strain suppresses the lifespan extension. While loss of *rde-4* did not further extend the lifespan of another insulin signaling mutant, *daf-2(e1370)*, the authors propose that this

may be due to the already decreased expression of *alg-3/4* in this mutant. The authors then endeavor to connect insulin signaling and endo-siRNA targets to the longevity effects. They observed an enrichment of major sperm protein (MSP) genes in the mis-regulated set for *age-1(hx546); rde-4(-)* compared to *age-1(hx546)*, which led them to examine an MSP receptor, *VAB-1*, as loss of *vab-1* was previously shown to cause an increase in the insulin signaling protein, *DAF-18*, leading to an extended lifespan. These analyses showed that loss of *alg-3/4* results in the up-regulation of some MSPs, mislocalization of *VAB-1*, and increased levels of *DAF-18* in oocytes. Overall, this study describes interesting new connections between small RNA pathways and insulin signaling for regulating animal longevity. Some of the major conclusions would be better supported with attention to the following:

1. The title states that sperm specific Argonautes (ie *ALG-3/4*) are activated by insulin/IGF-1 signaling. This conclusion seems to be based on microscopy of endogenously tagged GFP::*ALG-3* in insulin signaling mutant backgrounds, along with published RNAseq data comparing *alg-3* and *alg-4* mRNA levels in *daf-2(e1370)* versus WT. As *alg-3* and *alg-4* are transiently expressed during a short developmental period, microscopy and qRT-PCR analyses of their expression over a time course with properly staged animals seem important for supporting this key conclusion. The microscopy should include bright field images to assess the number and developmental consistency of the worms being analyzed for GFP expression. For example, delayed development of *daf-2(e1370)* might show delayed expression of *alg-3/4* but not really impaired expression of these genes.
2. The other part of the title "and non-autonomously promote aging" is not fully supported. The authors do show that loss of the sperm specific *alg-3* and *alg-4* argonautes leads to gene expression and protein localization changes in other tissues, but there is no direct evidence that these changes are important for the lifespan phenotypes. As this is likely difficult to experimentally demonstrate, the authors could just remove the mention of non-autonomous in the title and be cautious in their suggestion of this role throughout the manuscript.
3. Considering the prior, albeit variable, results indicating lifespan effects of *rde-4* and *rde-1* loss of function mutants opposite to those reported in this submission, are there independent loss of function alleles or rescue strains that can further support these unexpected opposite lifespan phenotypes?
4. I do not understand what the section about the *DAF-16::GFP* transgene adds to the paper. I understand the interest in analyzing *DAF-16* expression (and localization) but the strain used here seems to utilize a multicopy transgene that is vulnerable to silencing, which is expected to differ on the endo-siRNA pathway mutants being analyzed. Buried in this paragraph is the result that an endogenously tagged *DAF-16::GFP* was unaffected.
5. The section about *lea-1* regulation also seems unconnected to the longevity theme. I understand that it is part of the non-autonomous regulation argument but it is confusing and should either be better explained or removed.
6. Is there a direct connection between the MSP genes that go up or down and the differentially expressed endo-siRNAs in the *rde-4* mutants? As MSP genes also have a very short expression window during development, it is unclear if their mis-regulation is direct or a developmental timing affect. qRT-PCR in a time course could help address this issue.

Referee #3:

Liontis et al describe ageing effects of the *C. elegans* so-called 26G RNA pathway. They report interesting genetic interactions between the IIS pathway, that they should act via the transcription factor *DAF-16*, where *DAF-16* is proposed to repress *alg-3/4*. Finally, some cell-non-autonomous effects are proposed, where targets of *ALG-3/4*, which are expressed during spermatogenesis, have effects in the oocyte.

An effect of the 26G RNA pathway on ageing would be very interesting, as would be any insight into how these genes (*alg-3/4*) are regulated at the transcriptional level. However, the current manuscript unfortunately does not make a very convincing case. I believe there is interesting data contained in the paper, but its presentation makes it hard to distill a coherent picture of what is going on. As a result, the paper feels like a loose collection of data points. Somehow, these are all related to ageing but they are not brought together in a coherent model. Furthermore, the way some of the data is presented does not allow for a good assessment of experimental effects (notably RNAseq) and in some cases the wording is inaccurate. Overall, like it is, the paper is not a strong candidate for publication, because of a combination of convoluted presentation and incomplete analyses. That said, I do think that there is interesting data in this manuscript that is suitable for publication (notably the *DAF-16-ALG-3/4* link). However, this needs to be brought out much better, and needs to be deepened. Below, I list a number of more specific, experimental issues that will need to be addressed.

- 1 The RNA seq analysis is not well presented. Only Venn diagrams and table of up and downregulated gene are provided. This is not sufficient. To assess, the reader will need plots, not tables.
- 2 Many results are interpreted with the assumption that a small RNA matching to a gene implies direct regulation. As far as I am aware, *ALG-3/4* have not been shown to directly regulate any gene. Correlations have been published, but experiments testing direct importance of *ALG*-targeting are missing. Hence, many of the conclusions drawn in this work rely on assumptions, not facts. For instance, *lea-1* is presumed to be a direct target. However, no data exist to support this.
- 3 Also regarding *LEA-1* analyses: a cell-non-autonomous effect is claimed. This is based on the finding that defects in the oocyte are observed, while *ALG-3/4* should target *lea-1* during spermatogenesis. However, this can just as well be explained as an indirect effect. After all, the germlines of these nematodes first perform spermatogenesis (with *ALG-3/4*) and only during the last developmental stage switch to oogenesis. This spermatogenic phase should experience *LEA-1* dysregulation, and this could endure during later stages of life, in which oogenesis takes place. This is fundamentally different from cell-non-autonomous

effects, which implies communication of some sort between cells.

4 DAF-16 is proposed to repress alg-3/4. This would be a nice insight! However, the evidence is only indirect. More direct evidence would be needed to be convincing. This could be inducible overexpression of DAF-16, mapping of DAF-16 binding on the alg-3/4 genes etc. This could also make use of published CHIP data sets for instance. It does not necessarily involve a wet-lab experiment.

5 The section on the zls356 transgene is rather irrelevant. This seems to involve a transgene-RNAi effect, and not relevant to this paper.

6 The 'unnatural' death assay confused me. What is considered unnatural here, and what is the relevance to the main storyline?

7 The very first results section was unclear. Notably, the RNAseq section. We are presented with a number of correlations, but I lost the logic. I apologize for this somewhat vague comment, but I just cannot follow what was done why and what the conclusions are.

8 Page 3/4: 'there was further derepression...'. This sentence is supported by a Venn diagram, which does not report on extent of repression. They report just on how many genes do what. A different analysis is needed. This is in general a problem: the RNAseq data is interpreted mainly from such Venn diagrams, which takes out much of the quantification. Whether a gene is upregulated 2 fold or 100 fold, both would equally contribute to a Venn diagram.

9 Third paragraph page 4: the difference between synergistic and additive effects between mutants is not very obvious This needs quantification.

10 Supplemental Figure S1 is very subjective. Not sure how to fix this, but maybe some supplemental movies could underpin the findings.

11 How specific is the effect of age-1 on 22G-RNA-targeted genes? In other words, are non-22G targeted genes not affected at all?

12 Page 5, start of the DAF-16 section; first paragraph. Figures 3A/B do not show what is claimed. Maybe they do in a literal sense, but what is not shown is to what extent how many class 1 genes overlap with downregulated genes and vice versa. Hence, the analysis lacks a control, and is not really interpretable.

13 Some of the microscopy images are quantified. What was used for normalization? Also, better, higher resolution imaging is required to make the points; or perhaps the images are suitable, but they need to be presented in a much more zoomed-in manner

14 Is there any evidence that DAF-16 is regulated by ALG-3/4? I wonder if a regulatory feedback between DAF-16 and ALG-3/4 may be in place?

15 The authors repeatedly write about expression of ALG-3/4 and related factors in sperm. This is not correct. They are expressed during spermatogenesis. They are most likely not very abundant in sperm.

Referee #1:

With this manuscript, Lontis and colleagues aim to determine the role of endogenous small interfering RNAs in animal aging. Using the nematode *C. elegans* as a model, the authors uncover the interaction between components of the endo-siRNA pathway and insulin/IGF-1 signaling and provide key insights into how ALG-3/4 Argonautes modulate the lifespan of *C. elegans*.

Overall, this study is interesting and sheds light on the role of endogenous siRNAs in controlling animal aging. As detailed below, substantial work will be needed to strengthen the conclusion and make this work suitable for publication:

1-As ALG-3 and ALG-4 are primarily linked to male-specific siRNAs, it seems essential to assess their effects on the aging of male animals as well. The phenotypic and molecular impacts may be even more significant and biologically relevant for those animals. Moreover, these experiments will also contribute to refining the model related to the cell non-autonomous role of ALG-3 and ALG-4 in this process.

Thank you for bringing to our attention the idea of performing our assays in males. Lifespan experiments in males are complicated by their extremely prominent “escaping” behavior (over half of the male population throughout their lifespan), motivated by the search for a mate, and the negative effect from the presence of other males (Gems and Riddle 2000 and Shi, Runnel, Murphy 2017). We nevertheless now performed the male lifespan assay and controlled for this negative effect by keeping the number of males per plate constant (4/plate) throughout their lifespan, but note the variability of results between the three replicates (Dataset EV1). Ultimately, we did not observe the increased lifespan in age-1; alg-3; alg-4 triple mutant males (Fig. EV1E). The caveat is that we can only score males who do not escape from their plate, and there is a possibility that escapers might have lived longer.

Many molecular effects of daf-2 and alg-3; alg-4 were sustained in males (now Fig. EV4D, Appendix Fig. S1F). However, given the negative lifespan result and an absence of DAF-18 effects in males, we conclude that the communication between the spermatogenic and oogenic germlines in age-1; alg-3; alg-4 hermaphrodites is critical for the lifespan extension of this mutant strain.

2- Since the phenotypes observed in *hrde-1* and *nrde-3* mutants differ significantly from those of *alg-3* mutant animals, I am not convinced that they are the best genetic approach for studying the effect of ALG-3/4 on DAF-16 expression. The authors should instead use CRISPR-based genome editing to delete *alg-3* in the zls356 strain to

overcome the difficulty of genetic crossing and directly test the contribution of ALG-3/4 to DAF-16.

The effects of hrde-1 and nrde-3 mutants on zls356 are no longer included in this manuscript, per the recommendation of reviewers #2 and # 3

3- It appears that there are very few RDE-4-dependent small RNAs targeting the *lea-1* gene. Are they sufficient to affect its expression? The authors should monitor the expression of *Lea-1* mRNA to experimentally support its control by RDE-4.

Lea-1 was presumed to be a direct target positively regulated by ALG-3/4, based on a study using males (Conine et al., 2013). We agree that there are a few RDE-4-dependent sRNAs in hermaphrodites. Our experiments with LEA-1::GFP show that it is reduced in the distal germline and oocytes in alg-3/4(-). Thus, lea-1 may be largely indirectly regulated by RDE-4 and ALG-3/4 in hermaphrodites, which is consistent with our model of non-autonomous effects of ALG-3/4 on oocytes. Lea-1 may be directly regulated by ALG-3/4 in males.

4- Since the different mutants used in this study can affect germline integrity, it is important to include **DIC images** along with the fluorescent images found in **Supplemental Figure S7** to determine whether the observed effect on LEA-1 expression is due to morphological differences.

We now show both DIC and fluorescent images of LEA-1 expression in Appendix Fig S1.

5- It is quite challenging to determine any differences in VAB-1 localization in oocytes based on the images presented in Figure 6C. A set of images used to compile the data displayed in panels D and E should be included.

We now show confocal microscopy images in Fig. 6B, as well as a set of confocal images in Fig. EV5 illustrating the different phenotypes.

6- Even with some experience in the *C. elegans* small RNA field, I found this study rather complicated to follow. Careful editing will be necessary to make its key messages more accessible to readers.

We edited the manuscript to make it more accessible to the readers.

Referee #2:

This paper by Lionitis et al., explores roles for endogenous RNA interference (endo-RNAi) factors in regulating longevity in *C. elegans*. This is an important study because many pathways are known to influence lifespan in *C. elegans*, but few investigations have focused on the impact of endo-siRNA pathways in aging worms. While all tested loss of function mutants for endo-siRNA factors had minor, if any, lifespan phenotypes, they all seemed to enhance the extended lifespan of the insulin signaling mutant, *age-1(hx546)*, to various degrees (except *nrde-3*). *RDE-4* participates in the generation of most endo-siRNAs, but not miRNAs, and small RNA sequencing showed depletion of primary endo-siRNAs used by the *ALG-3/4*, *RDE-1* and *ERGO-1* Argonautes for regulating target mRNAs. In addition to loss of *alg-3/4* enhancing the extended lifespan of *age-1* mutants, this strain also exhibited prolonged healthspan compared to loss of *age-1* alone. Yet, fertility was compromised more than loss of *age-1* or *alg-3/4* on their own. Their RNA-seq data pointed to *DAF-16* activity as contributing to the enhanced lifespan resulting from loss of *rde-4* and *age-1* and, indeed, loss of *daf-16* in this strain suppresses the lifespan extension. While loss of *rde-4* did not further extend the lifespan of another insulin signaling mutant, *daf-2(e1370)*, the authors propose that this may be due to the already decreased expression of *alg-3/4* in this mutant. The authors then endeavor to connect insulin signaling and endo-siRNA targets to the longevity effects. They observed an enrichment of major sperm protein (MSP) genes in the mis-regulated set for *age-1(hx546); rde-4(-)* compared to *age-1(hx546)*, which led them to examine an MSP receptor, *VAB-1*, as loss of *vab-1* was previously shown to cause an increase in the insulin signaling protein, *DAF-18*, leading to an extended lifespan. These analyses showed that loss of *alg-3/4* results in the up-regulation of some MSPs, mislocalization of *VAB-1*, and increased levels of *DAF-18* in oocytes. Overall, this study describes interesting new connections between small RNA pathways and insulin signaling for regulating animal longevity. Some of the major conclusions would be better supported with attention to the following:

1. The title states that sperm specific Argonautes (ie *ALG-3/4*) are activated by insulin/IGF-1 signaling. This conclusion seems to be based on microscopy of endogenously tagged GFP::*ALG-3* in insulin signaling mutant backgrounds, along with published RNAseq data comparing *alg-3* and *alg-4* mRNA levels in *daf-2(e1370)* versus WT. As *alg-3* and *alg-4* are transiently expressed during a short developmental period, microscopy and qRT-PCR analyses of their expression over a time course with properly staged animals seem important for supporting this key conclusion. The microscopy should include bright-field images to assess the number and developmental consistency of the worms being analyzed for GFP expression. For example, delayed development of *daf-2(e1370)* might show delayed expression of *alg-3/4* but not really impaired expression of these genes.

We agree that the delayed development of daf-2(e1370) is an important consideration. We repeated the experiments in a time course with young L4, moderate L4, late L4, young adults, moderate Day 1 adults (near the timing of the presence of the first embryo), and Day 2 (D2) adults (24 hours after the young adult stage). The animals were staged based on vulva development (shown on DIC images) and used for RT-qPCR and GFP::ALG-3 imaging in Fig. 4 and EV4A,B. In daf-2(-) mutants, there is initially a DAF-16-independent decrease in alg-3 and alg-4, likely due to the slower growth of daf-2(-) and daf-2(-); daf-16(-), which may have been accompanied by a delay in spermatogenesis relative to vulval development. In late L4, there is a DAF-16-dependent decrease in alg-3/4 RNA and a corresponding decrease in ALG-3 protein and an increase in ALG-3/4 target MSP mRNA expression (Fig. 5D–F). In young adults, the alg-3/4 RNA has quickly decreased in all strains except daf-2(-), where the decrease is delayed to Day 2 adults. However, the DAF-16-dependent decrease of ALG-3 protein in daf-2(-) persists in young adults, probably due to the lag between transcription and translation.

2. The other part of the title "and non-autonomously promote aging" is not fully supported. The authors do show that loss of the sperm specific alg-3 and alg-4 argonautes leads to gene expression and protein localization changes in other tissues, but there is no direct evidence that these changes are important for the lifespan phenotypes. As this is likely difficult to experimentally demonstrate, the authors could just remove the mention of non-autonomous in the title and be cautious in their suggestion of this role throughout the manuscript.

We changed the title to “Mutual regulation of spermatogenesis-specific Argonaute proteins and insulin/IGF-1 signaling in aging control” to reflect our findings better and used caution to separate the description of “non-autonomous effects” from “non-autonomous effect on aging” in the text.

3. Considering the prior, albeit variable, results indicating lifespan effects of rde-4 and rde-1 loss of function mutants opposite to those reported in this submission, are there independent loss of function alleles or rescue strains that can further support these unexpected opposite lifespan phenotypes?

The lifespan of rde-4 mutants is inherently variable. We conclude this based on the variability of 8 individual replicates of the lifespan experiment with the same outcrossed rde-4(-) strain. Since RDE-4 is involved in both ERGO-1-bound and ALG-3/4-bound sRNA production, the opposite effects of these pathways on lifespan likely underlie the variability of the mutant phenotype. The short-lived

phenotype of nrde-3(-) shown in this manuscript represents the ERGO-1 pathway. Notably, the effects of rde-4 and rde-1 mutants on lifespan are small compared to the effects we see with rde-4(-) and alg-3/4(-) combinations with age-1 reduction-of-function. Our manuscript is focused on the latter.

4. I do not understand what the section about the DAF-16::GFP transgene adds to the paper. I understand the interest in analyzing DAF-16 expression (and localization) but the strain used here seems to utilize a multicopy transgene that is vulnerable to silencing, which is expected to differ on the endo-siRNA pathway mutants being analyzed. Buried in this paragraph is the result that an endogenously tagged DAF-16::GFP was unaffected.

We removed the DAF-16::GFP transgene data from the manuscript; they will be described separately.

5. The section about lea-1 regulation also seems unconnected to the longevity theme. I understand that it is part of the non-autonomous regulation argument but it is confusing and should either be better explained or removed.

Yes, the lea-1 regulation is part of the non-autonomous regulation argument. We tried to better integrate it in the revised version of the manuscript.

6. Is there a direct connection between the MSP genes that go up or down and the differentially expressed endo-siRNAs in the rde-4 mutants?

Thank you for this insightful question. The MSPs that are increased in age-1; rde-4 mutants have corresponding decreases in endo-siRNAs. There appears to be an anticorrelation between the downregulated endo-siRNAs and upregulated mRNA of these MSP genes ($r = -0.39$). However, because there are only 10 upregulated MSP genes, this correlation coefficient is not statistically significant ($p = 0.39$). On the other hand, the more numerous MSP genes downregulated in age-1; rde-4 mutants correlate with decreases in endo-siRNAs ($r = 0.83$, $p=0.01$). We attribute this correlation to overall defects in spermatocyte function caused by the rde-4 mutation (Fig. 2E). Because the statistical tests are underpowered for correlations between MSP mRNA and siRNA changes, we do not report these correlations in the manuscript.

As MSP genes also have a very short expression window during development, it is unclear if their mis-regulation is direct or a developmental timing affect. qRT-PCR in a time course could help address this issue.

Based on the time-course experiments, the upregulation of the most highly expressed MSP genes is coincidental with alg-3/4 upregulation, now in Fig. EV4A,B, and Fig. 5D–F.

Moreover, we now cite evidence correlating the key MSP gene expression changes and lifespan phenotypes of established long-lived and short-lived mutants. Notably, downregulation of at least one MSP gene, dct-9, suppressed the enhanced lifespan of daf-2 mutants (Pinkston-Gosse & Kenyon, 2007).

Referee #3:

Liontis et al describe ageing effects of the C. elegans so-called 26G RNA pathway. They report interesting genetic interactions between the IIS pathway, that they should act via the transcription factor DAF-16, where DAF-16 is proposed to repress alg-3/4. Finally, some cell-non-autonomous effects are proposed, where targets of ALG-3/4, which are expressed during spermatogenesis, have effects in the oocyte.

An effect of the 26G RNA pathway on ageing would be very interesting, as would be any insight into how these genes (alg-3/4) are regulated at the transcriptional level. However, the current manuscript unfortunately does not make a very convincing case. I believe there is interesting data contained in the paper, but its presentation makes it hard to distill a coherent picture of what is going on. As a result, the paper feels like a loose collection of data points. Somehow, these are all related to ageing but they are not brought together in a coherent model. Furthermore, the way some of the data is presented does not allow for a good assessment of experimental effects (notably RNAseq) and in some cases the wording is inaccurate. Overall, like it is, the paper is not a strong candidate for publication, because of a combination of convoluted presentation and incomplete analyses. That said, I do think that there is interesting data in this manuscript that is suitable for publication (notably the DAF-16-ALG-3/4 link). However, this needs to be brought out much better, and needs to be deepened. Below, I list a number of more specific, experimental issues that will need to be addressed.

1 The RNA seq analysis is not well presented. Only Venn diagrams and table of up and downregulated gene are provided. This is not sufficient. To assess, the reader will need plots, not tables.

We agree that quantifiable plots add valuable information. We now present RNA-seq data in scatter plots and heat maps in Fig. 1C–G.

2 Many results are interpreted with the assumption that a small RNA matching to a gene implies direct regulation.

We (and others in the field) assume direct regulation based on reduced sRNA abundance (and changed mRNA expression, if data are available) in a specific RNAi mutant. This does not exclude the additional indirect effects of RNAi mutations on the expression of a specific gene. Note that we used published datasets for AGO target definition, which are widely accepted in the field.

As far as I am aware, ALG-3/4 have not been shown to directly regulate any gene. Correlations have been published, but experiments testing the direct importance of ALG-targeting are missing.

Again, it is rare for the field to show sRNA-dependent AGO binding to mRNA/pre-mRNA targets; thus, the assumptions are made as stated above. For spermiogenesis-specific and low-expressed ALG-3/4, CLIP experiments would be technically challenging.

Hence, many of the conclusions drawn in this work rely on assumptions, not facts. For instance, *lea-1* is presumed to be a direct target. However, no data exist to support this.

Lea-1 was presumed to be a direct target in a study performed using *alg-3/4* mutant males (Conine et al., 2013). Our data argue that, in hermaphrodites, it may also be regulated indirectly.

3 Also regarding LEA-1 analyses: a cell-non-autonomous effect is claimed. This is based on the finding that defects in the oocyte are observed, while ALG-3/4 should target *lea-1* during spermatogenesis. However, this can just as well be explained as an indirect effect. After all, the germlines of these nematodes first perform spermatogenesis (with ALG-3/4) and only during the last developmental stage switch to oogenesis. This spermatogenic phase should experience LEA-1 dysregulation, and this could endure during later stages of life, in which oogenesis takes place. This is fundamentally different from cell-non-autonomous effects, which implies communication of some sort between cells.

We agree that the effect on LEA-1 could be indirect. However, this effect is likely cell-non-autonomous because *alg-3/4* mutants also show strongly decreased LEA-1 levels in the distal mitotic cells, which include pachytene germline stem cells. These distal mitotic cells do not express ALG-3/4 and precede the

spermatogenic germline region. Therefore, the decrease in LEA-1 in these cells cannot be explained by enduring effects from the loss of ALG-3/4 from the spermatogenic phase.

4 DAF-16 is proposed to repress alg-3/4. This would be a nice insight! However, the evidence is only indirect. More direct evidence would be needed to be convincing. This could be inducible overexpression of DAF-16, mapping of DAF-16 binding on the alg-3/4 genes etc. This could also make use of published CHIP data sets for instance. It does not necessarily involve a wet-lab experiment.

We observe a DAF-16-dependent repression of GFP::ALG-3 protein in young adult hermaphrodites, and a less prominent DAF-16-dependent repression of alg-3 and alg-4 mRNAs at the late L4 stage of daf-2 mutant worms. There could be both direct and indirect effects of DAF-16 on alg-3/4 transcription and ALG-3 protein abundance. We argue that this regulation is functionally significant for IIS mutant lifespan, and especially for the synergy between the RNAi mutants (rde-4, alg-3/4) and the weak age-1 mutant. We believe that functional significance is more important than proving a direct DAF-16-dependent repression of alg-3/4. The CHIP data for DAF-16 are available, but they were not performed on purified late L4 germlines or with tagged DAF-16 expressed exclusively in the germline. Moreover, transcription factor binding does not prove regulation, only suggests it.

Figure for referee with unpublished data and its description has been removed upon request by the authors.

5 The section on the zls356 transgene is rather irrelevant. This seems to involve a transgene-RNAi effect, and not relevant to this paper.

We removed the DAF-16::GFP transgene data from the manuscript; they will be described separately.

6 The 'unnatural' death assay confused me. What is considered unnatural here, and what is the relevance to the main storyline?

We agree that this section could be better explained. In the C. elegans aging field, deaths are considered “natural” when an animal gradually ceases to move over several days, and considered “unnatural” or “accidental” when an incident occurs to quickly cause death, e.g. crawling to the side of the plate where worms dry out, internal hatching of embryos causing a “bagged” phenotype, vulval bursting, and intestine leaking out of the burst vulva (which tend to cause death with a day or so of the phenotype onset). These accidental deaths are commonly right-censored from the lifespan data forming lifespan curves (Park et al, 2017; Cornwell & Samuelson, 2022), but we believe they can provide important insights into the health and physiology of the animals.

For example, while alg-3; alg-4 worms lived slightly longer than WT (+5% mean lifespan) according to their natural deaths of aging, their population showed a high level of intestine leaking (25% of worms compared to 10% in WT, Fig. EV2D). This suggests the alg-3/4 mutations are not beneficial in terms of overall life expectancy. It was striking, however, that the intestine-leaking phenotype caused by alg-3/4 mutations was suppressed in the age-1 mutant background (Fig. EV2D).

We clarified the relevance of these assays in the manuscript.

7 The very first results section was unclear. Notably, the RNAseq section. We are presented with a number of correlations, but I lost the logic. I apologize for this somewhat vague comment, but I just cannot follow what was done why and what the conclusions are.

We apologize for failing to concentrate on the most prominent connection between the rde-4 and age-1 mutant RNA sequencing data and the ALG-3/4 pathway. Other correlations may or may not be functionally relevant. We made corresponding changes in the revised manuscript.

8 Page 3/4: 'there was further derepression...'. This sentence is supported by a Venn diagram, which does not report on extent of repression. They report just on how many genes do what. A different analysis is needed. This is in general a problem: the RNAseq data is interpreted mainly from such Venn diagrams, which takes out much of the quantification. Whether a gene is upregulated 2 fold or 100 fold, both would equally contribute to a Venn diagram.

To address this point, we now show quantitative data of gene expression changes using heat maps in Fig. 1G.

9 Third paragraph page 4: the difference between synergistic and additive effects between mutants is not very obvious. This needs quantification.

The quantification is now included in the text.

10 Supplemental Figure S1 is very subjective. Not sure how to fix this, but maybe some supplemental movies could underpin the findings.

To control for subjectivity, the scoring of the strains was done blindly, and the genotype was assigned to the data after the lifespan experiment was completed. The presented experiments took months, if not years, to complete; we cannot repeat them for filming.

Nevertheless, we removed daf-2 and daf-2; rde-4 qualitative spontaneous movement data because it is not supported by separate quantitative data. We kept age-1 and age-1; alg-3; alg-4 qualitative data (Fig. EV1G) because it is supported by the quantitative data in Fig. 2B.

11 How specific is the effect of age-1 on 22G-RNA-targeted genes? In other words, are non-22G targeted genes not affected at all?

Most, if not all, C. elegans genes have some 22G-RNAs matching them according to recent studies. We (and others in the field) assume direct regulation (targeting) based on reduced sRNA abundance (and changed mRNA expression, if data are available) in a specific RNAi mutant. Our data show that genes known to be targeted by RDE-4- and ALG-3/4-dependent sRNAs are misregulated in the age-1 mutant. However, the majority of genes changing in the age-1 mutant are not designated as Argonaute targets, i.e., they are technically non-22G targeted.

12 Page 5, start of the DAF-16 section; first paragraph. Figures 3A/B do not show what is claimed. Maybe they do in a literal sense, but what is not shown is to what extent how many class 1 genes overlap with downregulated genes and vice versa. Hence, the analysis lacks a control, and is not really interpretable.

We had performed these controls separately as sanity checks, and indeed age-1(-) upregulated genes do not significantly overlap with Class 2 genes, and downregulated genes with Class 1 genes, etc. However, we moved the focus of our manuscript away from DAF-16 targets, and we therefore removed these overlaps as well.

13 Some of the microscopy images are quantified. What was used for normalization? Also, better, higher resolution imaging is required to make the points; or perhaps the images are suitable, but they need to be presented in a much more zoomed-in manner.

We now clarify our methods for normalization in the Methods section: “Controls (e.g., WT) were always present on the exact same surface (same slide, agarose pad, and coverslip) as experimental animals (e.g. mutants) subjected to the same exposure. The experimental signal was normalized to controls in each replicate, and results were pooled for graphs presented.”

We agree that zooming would add clarity to our figures and have now cropped/zoomed images throughout the figures.

We now present new, high-resolution, confocal images of VAB-1::GFP and its localization changes in the mutants in Fig. 6B and Fig. EV5.

14 Is there any evidence that DAF-16 is regulated by ALG-3/4? I wonder if a regulatory feedback between DAF-16 and ALG-3/4 may be in place?

No evidence of direct regulation: we have not observed changes of endogenously tagged DAF-16::GFP in *alg-3/4(-)* or *rde-4(-)* conditions, although DAF-16 localization dynamics in this strain are extremely fast (on the order of 1-4 minutes), making it challenging to observe small changes in nuclear localization. However, our model connects the ALG-3/4 function with the Eph receptor (VAB-1) and PTEN (DAF-18) through MSP signaling (ALG-3/4 represses lifespan-promoting MSP genes). DAF-18, in turn, regulates DAF-16 activity and lifespan.

15 The authors repeatedly write about an expression of ALG-3/4 and related factors in sperm. This is not correct. They are expressed during spermatogenesis. They are most likely not very abundant in sperm.

Thank you for this correction. This mistake has been corrected in the revised manuscript.

21st Oct 2025

Dear Dr. Grishok,

Thank you for the submission of your revised manuscript. We have now received the comments from all referees as well as cross-comments from referee 1. Referee 2 was unfortunately not available for cross-comments. I am sorry to say that the evaluation of your revised ms is not a positive one.

As you will see, while referee 2 is more positive and all referees acknowledge that several points have been addressed, both referees 1 and 3 do not find the data sufficiently strong and convincing for publication by EMBO reports. I read your revised ms again and am sorry to say that I agree with this assessment. I also re-discussed your study with my colleagues here, and the outcome of these discussions is that we can unfortunately not offer to publish it. For EMBO reports, better, more direct evidence for how ALG-3 and ALG-4 impact lifespan would be required.

While we cannot pursue this manuscript further, we encourage you to transfer your study to our not-for-profit open-access sister journal, Life Science Alliance (LSA). We shared your manuscript and the accompanying reviews with LSA Executive Editor, Tim Fessenden, who is interested in your findings. He is pleased to offer publication of your manuscript at LSA without further revision. You may use the link below to immediately transfer your manuscript to LSA. We encourage you to contact Dr. Fessenden at t.fessenden@life-science-alliance.org to discuss this work or to address any questions you may have.

I am sorry that I cannot be more positive for EMBO reports and hope that you view the possibility of transferring your ms to LSA favourably.

Kind regards,
Esther

Referee #1:

The authors made a considerable effort to address my comments and concerns regarding the manuscript. Although it would have been helpful to obtain more robust experimental data to directly support the role of ALG-3/4 in male aging and the regulation of *Lea-1*, I understand the limitations in acquiring such data. Nevertheless, I agree that the revised manuscript is improved and provides interesting insights into how siRNAs contribute to the aging process, which will interest the community and warrants its acceptance for publication. However, I am unsure whether this is sufficient overall for publication in EMBO Reports.

Referee #2:

The authors have satisfactorily answered my questions in the revised version of this paper.

Referee #3:

The authors addressed many of the points raised and the manuscript became better readable. I still have my doubts about how direct all these findings are. This is not per-se a shortcoming of the authors, however, as whether specific genes are directly targeted (meaning silenced) by 22G RNAs is just not very clear. The authors are right when they comment in the rebuttal that in 'the field' this is commonly done, but that does not make it right. The unaware reader should be aware what the proof is that (for instance) *lea-1* is directly silenced by 22G RNA. I think there is no evidence other than correlations and assumptions. In fact, the authors show now that indirect effects may be more relevant.

Another remaining issue I have relates to the supposed cell-non-autonomy:

Comment from authors in my review comment:

We agree that the effect on *LEA-1* could be indirect. However, this effect is likely cell-non-autonomous because *alg-3/4* mutants also show strongly decreased *LEA-1* levels in the distal mitotic cells, which include pachytene germline stem cells. These distal mitotic cells do not express *ALG-3/4* and precede the

spermatogenic germline region. Therefore, the decrease in LEA-1 in these cells cannot be explained by enduring effects from the loss of ALG-3/4 from the spermatogenic phase.

Reply:

1) Distal mitotic cells cannot be in pachytene (=meiosis)

2) Maybe I did not make my point clear. I understand that alg-3/4 are not expressed in these cells at that stage, however, at earlier developmental time these cells did express alg-3/4 and they could affect the future gene expression of these cells, like regularly happens during development. This is not the same as cell-non-autonomous; rather it could be more like a developmental program.

All in all I am luke warm about the suitability of this work for EMBO Rep as the causative links remain rather unclear.

Cross-comments from referee 1:

The Reviewer 3 raised valid points that align with mine. I was also on the fence with this. Perhaps LSA would be a more suitable journal for this work.

** As a service to authors, EMBO Press provides authors with the ability to transfer a manuscript that one journal cannot offer to publish to another journal, without the author having to upload the manuscript data again. To transfer your manuscript to another EMBO Press journal using this service, please click on Link Not Available

Dear Esther,

I received your decision letter regarding EMBOR-2025-61838V2 and I am very puzzled by it.

You have agreed with a revision plan for EMBOR-2025-61838V1 per the communication below, and we have fulfilled the plan by executing additional experiments, providing additional analyses, re-writing the paper, etc.

The quality of re-reviews by reviewers #1 and #3 is very poor, with no relevant specific comments. They both largely comment on the regulation of the LEA-1 gene which is now presented in the appendix and has no relevance to our main findings and models. Would removing LEA-1 from the paper address the concerns? This is easy to do.

I am not sure what part of the reviewers' comments you agree with. Please explain. I provide a comprehensive rebuttal of the comments below.

I could have accepted your view that the paper is not up to EMBO Rep standards, if you have not sent it to review twice. You did not use this argument when we first submitted the manuscript. Now, the paper is improved and accepted by two reviewers who do not question the quality of research (although reviewer #1 has some poorly explained subjective negative feelings). Therefore, I would like you to provide us with specific reasons.

As a peer reviewer, I give very specific and constructive criticism to the authors and articulate my arguments well. I expect the same quality of assessment of my work.

I hope that you'll look at the revised manuscript and my rebuttal of the new "reviews" and consider changing your decision. If a more formal appeal is required, please let me know how to proceed.

Thank you,

Alla

Rebuttal:

Reviewer #1

"The authors made a considerable effort to address my comments and concerns regarding the manuscript. Although it would have been helpful to obtain more robust experimental data to directly support the role of ALG-3/4 in male aging and the regulation of Lea-1, I understand the limitations in acquiring such data. Nevertheless, I agree that the revised manuscript is improved and provides interesting insights into how siRNAs contribute to the aging process, which will interest the community and **warrants its acceptance for publication**. However, I am unsure whether this is sufficient overall for publication in EMBO Reports."

- "obtain more robust experimental data to directly support the role of ALG-3/4 in male aging". Our new findings demonstrate that ALG-3/4 **do not** regulate male aging, only hermaphrodite aging. Does the reviewer mean that we **should** see the effect in males? This could be a hermaphrodite-specific role of ALG-3/4 in aging through nonautonomous effects on oogenic germline.
- "and the regulation of Lea-1". Lea-1 regulation is not related to the regulation of aging by ALG-3/4. It is peripheral to the main findings and is now included in the appendix. This can be removed from the manuscript.
- "I am unsure whether **this** is sufficient overall for publication in EMBO Reports". What is "this" and why is it not sufficient? Does the reviewer specifically want to see the effect of ALG-3/4 on aging in males through LEA-1? Why is such hypothetical scenario more interesting than our model? These types of arguments are very subjective and have nothing to do with merit but rather with personal preferences of scientists/reviewers.

Reviewer #3

"The authors addressed many of the points raised and the manuscript became better readable. I still have my doubts about how direct all these findings are. This is not per se a shortcoming of the authors, however, as whether specific genes are directly targeted (meaning silenced) by 22G RNAs is just not very clear. The authors are right when they comment in the rebuttal that in 'the field' this is commonly done, but that does not make it right.

The unaware reader should be aware what the proof is that (for instance) lea-1 is directly silenced by 22G RNA. I think there is no evidence other than correlations and assumptions. In fact, the authors show now that indirect effects may be more relevant."

- Lea-1 regulation is irrelevant for the main logic of the story, it can be removed.
- Our findings describe **indirect positive** regulation of lea-1 by ALG-3/4, not its **direct silencing**.
- "what the proof is that (for instance) lea-1 is directly silenced by 22G RNA". We make no claims of direct silencing of lea-1 by siRNAs. Direct lea-1 activation, not silencing, by 22G RNA in males was claimed by Mello and co-workers in 2013 Cell publication. If the reviewer wants to dispute direct regulation of lea-1 or other target genes by ALG-3/4-dependent 22G RNAs, they should correspond with Cell and/or Craig Mello directly.
- Again, elevated expression of mRNA and reduced expression of antisense siRNAs specific to them in mutants of Argonaute proteins that were shown to exist in an immunoprecipitated complex with such siRNAs is the standard in the field. If the reviewer wants to change the standards, they should bring this up at scientific meetings. Moreover, the reviewer does not indicate what additional proof they want to see. We currently claim that silencing of several msp genes by ALG-3/4 (namely, ssp-16, melo-3, and dct-9) restricts lifespan in *C. elegans*. All these genes were described as ALG-3/4 targets by Mello and colleagues in 2010 and subsequent publications by others. In fact, ssp-16 is THE MODEL target of ALG-3/4 often chosen for illustration of genomic data.
- If the Editors of EMBO rep, together with the reviewer, question published data and standards in the field, I want to have a written statement to that effect. I provide a list of *C. elegans* RNAi publications in EMBO rep where no mechanistic insights about the processes were provided. Often, not even elevation of target mRNA expression together with reduced antisense siRNA expression (or

reduction in mRNA expression correlated with siRNA increase) is shown to claim regulation by RNAi.

<https://www.embopress.org/doi/full/10.1038/s44319-025-00512-7>

The focus is RNAi-based gene (reporter) regulation. No mRNA or siRNA data are provided because of sufficient knowledge in the field. However, the effects seen on reporters in RNAi mutants could potentially be due to indirect effects, such as known regulation of endogenous RNAi processes by studied factors, which may have secondary effects on transgene expression.

<https://www.embopress.org/doi/full/10.1038/s44319-025-00543-0>

Largely phenotypic data, no siRNA data shown that would correlate with mRNA target changes shown by RT-qPCR. No rigorous data to fully and mechanistically support the speculative final models.

<https://www.embopress.org/doi/full/10.15252/embr.202357250>

A mechanistic study of miRNA-dependent gene silencing. Only mRNA levels of known miRNA targets are assayed by RT-qPCR in specific conditions, including when specific genes are inhibited by RNAi. No validation of RNAi efficiency in silencing of target genes is shown by either western, IF, or RT-qPCR. Possible indirect effects are not discussed.

We show RNA-seq, siRNA-seq and time-course RT-qPCR data for established endogenous ALG-3/4 targets *ssp-16*, *melo-3*, and *dct-9* in our paper. Yet, ALG-3/4-dependent regulation of such published targets is being questioned!

- siRNAs are routinely used as tools to silence genes in cell culture and reduction in mRNA expression is an accepted outcome of their efficiency, despite the possibility indirect effects. The list of such papers published in EMBO Rep will be enormous.

"Another remaining issue I have relates to the supposed cell-non-autonomy:

Comment from authors in my review comment:

We agree that the effect on LEA-1 could be indirect. However, this effect is likely cell-non-autonomous because *alg-3/4* mutants also show strongly decreased LEA-1 levels in the distal mitotic cells, which include pachytene germline stem cells. These distal

mitotic cells do not express ALG-3/4 and precede the spermatogenic germline region. Therefore, the decrease in LEA-1 in these cells cannot be explained by enduring effects from the loss of ALG-3/4 from the spermatogenic phase.

Reply:

1) Distal mitotic cells cannot be in pachytene (=meiosis)

2) Maybe I did not make my point clear. I understand that alg-3/4 are not expressed in these cells at that stage, however, at earlier developmental time these cells did express alg-3/4 and they could affect the future gene expression of these cells, like regularly happens during development. This is not the same as cell-non-autonomous; rather it could be more like a developmental program."

- This argument about LEA-1 regulation is blown out of proportion and not related to the main story. We show nonautonomous regulation of VAB-1 and DAF-18, not LEA-1, by ALG-3/4 as a key longevity-relevant effect, which is consistent with all our data.

"All in all I am luke warm about the suitability of this work for EMBO Rep as the causative links remain rather unclear".

- This evaluation is vague. What is exactly not clear? The msp gene dct-9, which is 6-7-fold upregulated in alg-3/4 mutant, was implicated in lifespan control in a published work, through an unknown mechanism. We show that MSP upregulation, including dct-9, in alg-3/4(-) occurs at the same time as VAB-1 localization and DAF-18 expression are changed precisely as would be expected from elevation of secreted MSP (based on published literature). This provides a possible mechanism that would explain the opposite effects on lifespan by dct-9 and alg-3/4 mutants.

Alla Grishok, PhD

Associate Professor

Boston University

Chobanian & Avedisian School of Medicine

Department of Biochemistry & Cell Biology

Co-Director, BU Genome Science Institute

72 E Concord Street, K422

Boston, MA 02118

617-358-4525

agrishok@bu.edu

<https://www.bumc.bu.edu/biochemcellbio/profiles/alla-grishok/>

Dear Alla,

Thank you for the clarifying video chat last week. After lengthy discussions with the EMBO reports team and after talking to you, we have decided that we can offer to publish your ms following minor revisions. Please incorporate all changes that we discussed in the final ms file. There are also a few editorial requests that will need to be addressed before we can proceed with the official acceptance of your manuscript:

- Please add up to 5 keywords to the ms file.
- If the "National Institutes of Health Office of Research Infrastructure Programs (P40 OD010440)" is a separate funder from the other funders mentioned, it needs to be entered during online ms submission as a separate funder.
- Datasets EV4-EV6 need to be uploaded as separate/individual files and the legends need to be updated (Table S4-S6 is incorrect). The dataset legends need to be removed from the ms file.
- The 2 figures in the Appendix file can be uploaded as 2 more EV figures. You can have up to 8 EV figures. And the Appendix can then be deleted.
- Our systematic image analysis identified a possible reuse of cells between Figure 6B and Figure EV5 that is not listed in the legends. Can you please explain ?
- Please supply Figure EV3 and EV4 at a higher resolution. Currently the figure appears pixelated under analysis.

* Figure Legends - Comments *

- Please note that the exact p values are not provided in the legends of figures 1E, 5B, C; 7B, EV3 D, please provide exact values as reasonable.
- Please note that the error bars are not defined in the legends of figures 4B, C, D, E; 5B, C, D, E, F; 7A-C; EV5 A-E, please add.

I would like to suggest some minor changes to the abstract that needs to be written in present tense. Please let me know whether you agree with this:

The potential role of small interfering RNAs (siRNAs) produced from double-stranded RNA in aging has not been fully addressed. The networks of genes regulated by siRNAs and their partner Argonaute proteins are best understood in *C. elegans*, a pioneering model of aging and small RNA studies. By analyzing lifespans, gene expression, and siRNA populations in *C. elegans* mutants deficient in siRNA function and insulin/IGF-1 signaling (IIS), we discover that redundant spermatogenic Argonautes ALG-3 and ALG-4 are capable of regulating IIS in a non-cell-autonomous manner [Can cell-autonomous regulation be excluded? If not, please re-write, eg "likely non-cell..."], potentially through direct control of the Major Sperm Protein (MSP) genes in the germline [OK?]. MSPs and MSP domains of some mammalian proteins are secreted and directly inhibit the Eph receptor (EphR). In turn, EphR interacts with and inhibits the stability of PTEN, a major negative regulator of IIS. We show that enhanced MSP expression correlates with EphR mislocalization and elevated PTEN levels in oocytes of *alg-3/4(-)* worms. At the same time, ALG-3/4 expression is regulated by IIS [OK?]. Thus, we propose mutual non-cell-autonomous regulation [please confirm] of IIS and ALG-3/4 potentially through secreted ligands.

I also slightly modified the short summary. Please confirm whether a non-cell-autonomous regulation is the only valid interpretation based on your data:

Argonaute proteins ALG-3 and ALG-4 genetically interact with the Insulin/IGF-1 signaling pathway to promote aging. They non-cell-autonomously influence [alternatively "They impact EphR"] EphR localization and PTEN levels in oocytes, potentially through direct inhibition of secreted Major Sperm Protein genes.

Best regards,
Esther
Esther Schnapp, PhD
Senior Editor
EMBO reports

All editorial and formatting issues were resolved by the authors.

Dr. Alla Grishok
Boston University Chobanian & Avedisian School of Medicine
Biochemistry & Cell Biology
71 East Concord St., K422
Boston, MA 02118
United States

Dear Alla,

I am very pleased to accept your manuscript for publication in the next available issue of EMBO reports. Thank you for your contribution to our journal.

You may qualify for financial assistance for your publication charges - either via a Springer Nature fully open access agreement or an EMBO initiative. Check your eligibility: <https://link.springer.com/journal/44319/how-to-publish-with-us>

Best,
Esther

>>> Please note that it is EMBO Reports policy for the transcript of the editorial process (containing referee reports and your response letter) to be published as an online supplement to each paper. If you do NOT want this, you will need to inform the Editorial Office via email immediately. More information is available here: <https://link.springer.com/partners/embo-press/editorial-policies#Peer%20review>